# Hormesis and Ginseng: Ginseng Mixtures and Individual Constituents Commonly Display Hormesis Dose Responses, Especially for Neuroprotective Effects

**DOI:** 10.3390/molecules25112719

**Published:** 2020-06-11

**Authors:** Edward J. Calabrese

**Affiliations:** Department of Environmental Health Sciences, School of Public Health and Health Sciences, University of Massachusetts, Amherst, MA 01003, USA; edwardc@schoolph.umass.edu; Tel.: +1-413-545-3164

**Keywords:** hormesis, hormetic, ginseng, biphasic, neuroprotection, aging, Alzheimer’s Disease, Parkinson’s Disease, wound healing, preconditioning

## Abstract

This paper demonstrates that ginseng mixtures and individual ginseng chemical constituents commonly induce hormetic dose responses in numerous biological models for endpoints of biomedical and clinical relevance, typically providing a mechanistic framework. The principal focus of ginseng hormesis-related research has been directed toward enhancing neuroprotection against conditions such as Alzheimer’s and Parkinson’s Diseases, stroke damage, as well as enhancing spinal cord and peripheral neuronal damage repair and reducing pain. Ginseng was also shown to reduce symptoms of diabetes, prevent cardiovascular system damage, protect the kidney from toxicities due to immune suppressant drugs, and prevent corneal damage, amongst other examples. These findings complement similar hormetic-based chemoprotective reports for other widely used dietary-type supplements such as curcumin, ginkgo biloba, and green tea. These findings, which provide further support for the generality of the hormetic dose response in the biomedical literature, have potentially important public health and clinical implications.

## 1. Introduction

Ginseng is a widely consumed nutritional supplement with a long history of use in traditional Asian medicine for a range of conditions. Publications in the scientific literature on the pharmacology of ginseng were first listed in the Web of Science in 1905 on the pharmacology of ginseng [1]. While scientific research on ginseng was low through the first 70 years of the 20th century, there was a multidisciplinary resurgent interest in the biological properties of ginseng starting in the 1970s. In 1976, the Journal of *Ginseng Research* was started, reflecting progressive and broadening interest to the present. The range of publications has been highly varied, but with a strong biomedical orientation, with the general goal of assessing whether ginseng might enhance public health and medical prevention/treatment of a wide range of harmful conditions. The present paper provides the first integrative assessment concerning the capacity of ginseng mixtures and some of its specific constituents to induce hormetic dose responses by documenting their occurrence, generality, mechanistic basis, and potential biomedical significance.

Hormesis is a biphasic dose–response relationship that is characterized by low-dose stimulation and high-dose inhibition [2,3]. The magnitude of the low dose stimulation is modest with the maximum stimulation typically being in the 130–160% range (compared to control groups: 100%) [4]. The dose width of the low-dose stimulation is usually less than a 50-fold starting from the estimated toxic/pharmacological threshold (Figure 1). Hormetic responses may occur either by a direct stimulation or an overcompensation to a disruption in homeostasis/slight to modest toxicity. Preconditioning-mediated biological responses are examples of hormesis, displaying the typical hormetic dose response when sufficient conditioning doses are used in the experiment [5,6,7]. Hormetic dose responses are general, being independent of biological model, inducing agent, endpoint, level of biological organization, and mechanism [8]. A series of recent publications have documented hormetic dose responses in the biomedical literature for curcumin [9], Ginkgo biloba [10], and green tea [11]. The present findings show that ginseng-induced hormetic effects are also commonly reported that are broadly generalizable, affecting numerous organ systems, cell types, and endpoints, showing capacity to induce acquired resilience for various durations, within multiple and diverse experimental settings, with particular research focus directed toward neuroprotective effects (Table 1, Table 2 and Table 3). The paper assesses the published literature of individual ginseng constituents as well as various types of ginseng extract mixtures. The paper will also relate the hormetic dose response findings for ginseng to the broader hormetic literature and highlight their potential biomedical and clinical implications.

## 2. Search Strategy

PubMed, Web of Science, and Google Scholar databases were searched for articles using the terms “hormesis or hormetic and ginseng, and specific ginseng constituents (e.g., Rg1, Rb1, Rc, Rd, Re, ginseng mixtures, ginseng saponins, gintonin, polyacetylenes), dose response and ginseng; U-shaped dose response and ginseng; biphasic dose response and ginseng; preconditioning and ginseng; adaptive response and ginseng; stem cells and ginseng; ginseng and biphasic concentration response; ginseng and conditioning response”. All relevant articles were evaluated for the references cited and for all papers citing these papers. All research groups publishing ginseng dose–response relationships were assessed for possible relevant publications in the above databases.

## 3. Ginseng Constituent RG1

### 3.1. Rg1-Stem Cell Proliferation and Differentiation

That ginsenoside Rg1 could enhance the proliferation and differentiation of stem cells is a relatively new development. For example, Rg1 treatment enhanced neural progenitor cell (NPC) proliferation in hippocampal tissue [29], endothelial cells [30], and enhanced proliferation and neuro-phenotype differentiation of human adipose-derived stem cells (ASC) [31,32]. These studies, along with an hormetic dose response for NPC by Liu et al. [33] with Rg5 (Figure 2), provide the supportive publications for a framework for subsequent research described below to clarify the nature of the Rg1-mediated hormesis dose response, dose optimization, and underlying mechanisms of stem cell proliferation and differentiation.

Researchers from four stem cell areas have published hormetic dose responses with Rg1. In these studies, the authors assessed both cell proliferation and differentiation. With this dual endpoint consideration, hormetic dose responses were reported for cell proliferation for each of the four papers. Only Liang et al. [34] also showed hormetic dose responses for differentiation endpoints. In the remaining three papers, differentiation endpoints had only one dose, precluding a hormetic dose response evaluation. The four stem cell publications involved the following study subjects: Adipose Stem Cells/ASC: a 23-year-old female [34], neural stem cells: Sprague–Dawley rats—four pregnant females (E-17), with the number of fetuses not reported (Figure 3) [35]; human derived pulp stem cells—50 third impacted molars from subjects 19–28 years of age (gender not mentioned) [36]; human periodontic ligament stem cells, with 10 adults (gender not given) [37]. Each study differed in the number of concentrations tested, ranging from a low of four concentrations with two concentrations below the threshold (ASC) [34] to a high of 10 concentrations with five concentrations below the threshold (NSC) [35].

Despite these study differences, each displayed consistent evidence of hormetic dose responses (Figure 2, Figure 3, Figure 4, Figure 5, Figure 6 and Figure 7). The maximum stimulation responses ranged from 130–250% while the stimulation concentration ranged from about 2 [36] to 100-fold [37]. The quantitative features of stem cell proliferation (Figure 4), stem cell paracrine activity (Figure 5a,b), and differentiation responses (e.g., adipocytes/chondrocytres) (Figure 6) reflected the quantitative features of the hormetic dose response [34]. The implications of hormetic–biphasic dose responses of Rg1 for the different stem cells remains to be explored. However, Yin et al. [37] suggests that Rg1 be further assessed as a possible medication for osteogenesis treatment since it enhances the proliferation of human periodontal ligament stem cells in an hormetic-like biphasic dose–response manner (Figure 7). Showing similar biphasic dose–response features, Wang et al. [36] suggested that Rg1 has potential as a pulp-capping agent to enhance pulp healing and preserve pulp vitality. Furthermore, porcine blastocyte hormetic findings were consistent with the above reports as Rg1 mediated protection via decreasing oxidative stress and increasing the cellular uptake of glucose that reduced apoptosis-induced cell death [38]. Finally, additional neural stem cell/hormesis findings are presented on neonatal brain hypoxia within the Rg1 neuroprotection section [39].

### 3.2. Rg1 Neuroprotection

Rg1 affects neuroprotection responses across a spectrum of neurological diseases and injury/damage conditions. These include Parkinson’s and Alzheimer’s Diseases, stroke, neonatal brain hypoxia-induced injury, and recovery/healing following spinal cord and peripheral nerve injuries and pain. The research approach has emphasized a combination of pre- and post-conditioning experimental protocols, co-administration of the protective/harmful agents, as well as the direct stimulation of neuronal tissue to enhance neuronal survival, neurite outgrowth, and cell migration activities in neuronal wound-healing studies. Furthermore, most of these investigations incorporate mechanistic features that clarified aspects of neuroprotective pathway components and their functions.

### 3.3. Parkinson’s Disease

#### 3.3.1. Introduction

In the case of Parkinson’s Disease (PD) and Rg1, three experimental models have been employed, including one in vivo mouse model study [40] and two in vitro models using the SK-N-SH (human neuroblastoma cell line that can express dopamine markers) [41] and the MES23.5 cell line (i.e., a dopaminergic cell line) [42], with each using a preconditioning protocol. Furthermore, the PD symptoms/endpoints were induced by standard/classical chemical agents. In the case of the in vitro experiments, 6-OHDA [41,42] and rotenone [43] were employed, whereas MPTP induced PD symptoms in the in vivo mouse study [40].

#### 3.3.2. In Vitro Studies

In the in vitro studies, the SK-N-SH cell line investigations [41,43] employed a one-hour pretreatment while the MES23.5 cell line study [42] used a 24 h pretreatment. Two in vitro studies [41,42] used the same 6-ODHA concentration of 100 μM with one administering it for 24 h (i.e., SK-N-SH), while the other did so for 48 h (i.e., MES23.5). In the case of the MES23.5 cells, there were 1000 cells/well that were grown for 24 h and then treated. In the SK-N-SH study of Gao et al. [41], cells were grown to 80–90% confluence prior to treatment. Since PD is less common in women than men, the thought that estrogen may protect the brain from developing PD has been widely discussed. Of relevance to the present assessment is that Rg1 was shown to be a phytoestrogen by Chan et al. [44] by enhancing the proliferation of MCF-7 cells in a hormetic manner. These hormetic findings were extended by Gao et al. [41], who reported that pretreatment with Rg1 prevented 6-ODHA induced neurotoxicity in SK-N-SH cells. This protection was blocked by the insulin-like growth factor 1-receptor (IGF-IR) antagonist JB-1 as well as by the estrogen receptor (ER) antagonist ICI 182780. Furthermore, a decrease in BCL-2 induced by 6-OHDA was reversed by the Rg1 pretreatment.

The findings indicated that the protective effects of Rg1 are mediated, at least in part, by its activation of the IGF-IR and ER signaling pathways. The study by Gao et al. [41] also revealed that 6-OHDA affected a downregulation of ER protein expression in SK-N-SH cells and that these effects were prevented by Rg1 preconditioning. Further, experimental evidence suggests that the ER and IGF-IR pathways interact, affecting multiple functions such as neuronal differentiation, plasticity, and neurodegeneration. While these findings support a conclusion that the protective effects of Rg1 on SK-N-SH cells depend on IGF-IR and ER, the research of Ge et al. [42] with MES23.5 cells indicated that Rg1 pretreatment mediated protection via the upregulation of BCL-2 gene expression, the activation of AKT phosphorylation, and the inhibition of ERK1/2 phosphorylation induced by 6-OHDA. However, Ge et al. [42] did not explore the role of IGF-IR and ER in the MES23.5 cell line research. Finally, the Fernandez-Mariano et al. [43] in vitro study, which used the SK-N-SH cell line with retonone as the stressor, also demonstrated a hormetic dose response. The mechanistic focus indicated a key role involving the upregulation of the NrF2 pathway that contributed to the neuroprotective response.

#### 3.3.3. In Vivo Studies

The in vivo (IP) mouse study of Chen et al. [40], which utilized a three-day preconditioning period, limited its hormetic-mediated neuroprotective effects to the enhanced formation of GSH in the substantia nigra and blockage of MPTP upregulation of JNK and cJUN at lower doses. The JNK signaling pathway mediates MPTP-induced neurotoxicity. These protective findings were also linked to increases in the BCL-2/BAX ratio, which estimates cellular ROS levels, providing a redox rheostat function.

#### 3.3.4. Summary

These findings indicate that Rg1 has the capacity to prevent the occurrence of Parkinson’s Disease-like processes in a hormetic dose–response manner via multiple complementary means in both in vitro and in vivo experimental protocols that both enhance adaptive capacities as well blunting agent-induced toxicities, with dose and the duration of exposure affecting the outcome.

### 3.4. Alzheimer’s Disease

Since Rg1 was used as a potential neuroprotective agent against PD, it is not surprising that it has also been applied to other neuronal diseases such as AD, since estrogen can reduce the production of beta-amyloid and diminish beta-amyloid-induced toxicity [45,46,47]. Using primary hippocampal neurons from the embryonic brain of Wistar rats (E-16-18) in a beta-amyloid toxicity study, Rg1 hormetically enhanced cell viability, reduced LDH release, and reduced apoptosis/caspase 3 activity [48]. Rg1 treatment also increased the BCL-2/BAX ratio, suggesting that the neuroprotective effect involves the regulation of caspase 3 levels. The authors raised the question of whether estrogen receptors may affect Rg1-induced neuroprotection. While these findings used an embryonic rat model for a disease of aging in human adults, the mechanistic concept is of potential significance. Subsequent research has supported these estrogen-mediated findings in models of AD using both mouse [49] and rat [50] studies.

### 3.5. Stroke

Publications linking stroke and Rg1-induced hormetic dose responses were framed differentially than those of PD/AD with no consideration given to its phytoestrogen properties either in hypothesis generation or mechanistic evaluation. In the case of stroke-Rg1 and hormesis, there were three papers, two using PC12 cells with either oxygen glucose deprivation (OGD)/Reperfusion (R) [51] (Figure 8) or CoCl2-inducing stroke-like symptoms/damage [52] and mouse NSC [53] and ODG. In the stroke model studies, the mouse NSC experimentation involved co-treatment with the Rg1 and OGD/R [54], whereas in the two PC12 cell studies, post-conditioning protocols were employed [51,52]. The concentration responses of each experiment were generally similar with optimal hormetic protective responses occurring in the 1–5 μM concentration range. However, the mechanistic experimental strategies involved considerably different approaches. In the case of the mouse NSC study [54], the Rg1 reduced OGD-induced cell damage by the inhibition of *p*-p38 and *p*-JNK2 expression, enhancing the BCL-2/BAX ratio and reducing caspase activities. The two PC12 cell studies extended the mechanistic involvement to include SIRT 1 activation and the inhibition of Toll-like receptor 4 (TLR-4)/myeloid differentiation factor 88 (MYD88) [40,51]. These two complementary processes affected the inhibitions of NF-kB transcription activity and the expression of multiple pro-inflammatory cytokines (e.g., IL-lB, TNFa, and IL-6). The anti-inflammatory effects also occurred with other ginsenosides but were more strongly seen with Rb1 and Rb3.

A third stroke approach explored the Rg1 neuroprotective mechanism by the MiR-144/NrF2/ARE pathway (Figure 8) [51]. The Rg1-induced protection was mediated by prolonging the nuclear accumulation of NrF2 and enhancing the expression of ARE target genes. The Rg1 effect was independent of dissociation of Keap-1 but involved post-translational processes. Further, Rg1 suppressed the expression of MiR-144 while increasing Nrf2 production. These findings indicate that Rg1 reduced oxidative stress after ischemic reperfusion (IR) via the inhibition of MiR-144 activity and the subsequent activation of the NRF2/ARE pathway at the post-translational level.

### 3.6. Neonatal Brain Hypoxia

Another neurological application of Rg1 treatment has involved protection against the hypoxia–ischemia encephalopathy in newborns and other age groups. In their study, Li et al. [39] reported that Rg1 biphasically enhanced the differentiation of NSC into neurons, stromal cells, and oligodendrocytes (Figure 9). In addition to optimizing the hormetic dose, they also placed the dose response within a temporal and developmental context. In these studies, developing neonatal rats receiving transplanted Rg1-induced NSCs at optimized hormetic doses displayed fewer pathological lesions along with significantly improved behavioral performance for multiple parameters (e.g., learning/memory endpoints). These findings are significant, since the Rg1 treatment not only affected the capacity to differentiate stem cells but also displayed enhanced functional intercellular communication capabilities.

## 4. Neuronal Wound Healing

### 4.1. Role of Schwann Cells

Several studies assessed the role of Rg1 on spinal cord and peripheral neurons [24,55] within an hormetic dose response context. The Liao et al. [24] experiments showed dose-dependent hormetic dose responses for the Sprague–Dawley rat spinal cord and neurons. Similar findings were reported for Rb1 and ginseng extracts, which are assessed in their respective sections and in the discussion section with a comparison figure. Using a commercial Schwann cell model (RSC96 cells), Lu et al. [55] reported that Rg1 induced hormetic biphasic dose responses for both cell viability and cell migration. Follow-up mechanistic efforts linked the enhanced cell proliferation to the upregulation of the IGF-1 and MAPK pathways and cell cycle proteins. The Rg1-enhanced cell migration activity was mediated by the FGF–2uPA–MMP-9 migration pathway. The respective pathway involvements were elucidated by specific pathway inhibitors. These findings suggest the potential utility of Rg1 to enhance neuron regeneration.

### 4.2. Role of Angiogenesis

An additional factor that may enhance the capacity for Rg1 to promote recovery from pathological conditions such as hypoxia/ischemia-induced brain damage, nerve injury, and/or myocardial infarction may be via the enhancement of angiogenesis, since it is a common factor of these diverse conditions. Rg1 is an effective inducer of angiogenesis involving cell migration and tubulogenesis via vascular endothelial growth factor receptor-2 (VEGFR-2) in studies with human umbilical vein endothelial cells (HUVEC) (Figure 10) [56]. The VEGFR-2 stimulation was mediated via the RG1 inhibition of MiR-15b expression. These findings suggest that Rg1 affects multiple MiRNAs and mediates different angiogenic mechanisms, including the production of angiogenic factors that enhance cell survival linked to angiogenesis.

### 4.3. Nerve Regeneration-Role of Schwann Cells

Ginseng has been employed to enhance neuronal protection in a range of experimental research contexts. In such studies, Rg1 displayed neuroprotective properties both reducing the magnitude of damage and facilitating the repair of nerve fiber injuries [57]. Follow-up studies revealed that Rg1 enhanced cultured Schwann cell proliferation and migration in response to peripheral nerve injuries [58,59]. The protective responses were mediated, in part, by the increased expression of glial-derived neurotrophic factor (GDNF) and BCL-2 [60]. Rg1 also enhanced repair and regeneration following sciatic nerve injury, restored the conductivity of regenerative fibers involved in limb motor function, and prevented skeletal muscle atrophy [61]. Follow-up investigations by Huo et al. [62] not only confirmed the findings of Zhou et al. [61] but extended the dose evaluation range, which revealed an hormetic biphasic dose response for nerve conduction velocity, the re-myelination of nerves, and nerve growth factor (NGF) neuronal expression.

### 4.4. Heart

Rg1 has also displayed hormetic dose responses in protecting embryonic rat heart-derived cells (H9cs) under conditions of nutrition-induced stress [63]. In the case of the cardiac nutritional stress, the experimental approach involved a preconditioning protocol in which the Rg1 was administered 12 h before the nutrition stress-induced injury, which involved incubation in a glucose-free medium. Cell viability was used as an indicator of the chemoprotective response. The Rg1 treatment rescued declining ATP levels and restored mitochondrial membrane potential. These effects were mediated via the activation of PTEN-induced kinase-1 (PINK 1), pAMPK, and aldolase signaling. Such actions of Rg1 are biomedically significant since they are part of an integrated mechanism to prevent mitochondrial damage, enhancing mitochondrial mediated autophagy and the regulation of mitochondrial fusion and fission.

### 4.5. Immune Responses

There has been surprisingly little hormesis-related research concerning the effects of Rg1 on immune responses. However, two relatively old papers have shown evidence of hormesis in this area. Both papers were unusual in that the 1980 paper of Tong and Chao [64] explored the capacity for Rg1 to induce a low-dose stimulation within the context of a circadian rhythm experimental framework. The hormesis findings were reported throughout the day at 11:00, 12:30, 14:00, and 15:30. In each case, the low-dose stimulation maximized at the same dose for each time period. The capacity of Rg1 to stimulate lymphocyte proliferation required the presence of mitogenic lectins such as phytohemagglutinin (PHA) or Con A. Thus, Rg1 was not able to stimulate proliferation in resting lymphocytes. Similar findings were later reported by Liu et al. [65] with lymphocytes from elderly subjects (65–78 years old) (Figure 11). In both studies, the Rg1 stimulation required the co-presence of PHA with the two agents acting in a synergistic fashion. In contrast to the stimulatory effects in the elderly, the joint exposure to young adults was inhibitory. These findings remain to be further explored.

### 4.6. Diabetes

In the STZ mouse diabetic model, Rg1 oral treatment reduced many diabetes-related conditions. The findings indicated reduced elevated blood glucose, diminished inflammatory factors such as Il-1B and Il-18, and decreased the NLRP3 inflammasome levels in the liver and pancreas [27]. Hormetic-like biphasic dose responses were reported for serum ALT/AST and insulin secretion with the responses being optimized at the intermediate dose.

These manifestations of inflammatory processes were blocked, at least, in part, via the upregulation of the NrF2/ARE pathways, decreasing the STZ induced inflammatory endpoint spectrum. It was also of interest that the Rg1 treatment decreased the capacity of STZ to affect the methylation of hepatic H3K9 in the liver and pancreas.

## 5. Ginseng Constituent RB1

### 5.1. Rb1: Neuroprotection

#### Alzheimer’s and Parkinson’s Diseases and Stroke Models

The ginsenoside Rb1 has been widely assessed for its neuroprotective features [66]. While most experiments used a limited number of doses, several papers reported hormetic–biphasic dose responses for Rb1. Of the eight in vitro hormetic studies, four utilized pre/post-conditioning and co-treatment experimental protocols. Of these preconditioning studies, two used cell lines (i.e., PC12 and SH-SY5Y), while the remainder employed embryonic Sprague–Dawley rat cortical tissue. The Rb1 preconditioning experiments used different preconditioning periods (1 h, 24 h) with a broad spectrum of stressor agents (i.e., Rotenone-SH-SY5Y cell line) (Figure 12) [43], AB-(25–35)- Sprague–Dawley cortical cells (Figure 13) [67], glutamine-Sprague–Dawley embryonic cortex cells (Figure 14) [68], CoCl2-PC 12 cells (Figure 15) [52] and tert-butylhydroperoxide (tBHT)-Sprague–Dawley embryonic neuroprogenitor cells (Figure 16) [69]. There was also considerable variation of stressor agent toxicity induction rates, ranging from a low of 35–45% (tBHT) [69] to 75–80% (CoCl2) [52]. Despite the range of model conditioning periods, stressor agents, and toxicity induction rates, each experimental setting displayed an hormetic–biphasic dose response with similar quantitative features. Furthermore, the magnitude of protection was also generally similar with optimal protection typically between 0.01 and 10 μM. Three of the preconditioning studies [43,52,69] provided relevant mechanistic findings for the Rb1-induced chemoprotection. In the cases of the rat NPC [69] and SY-SH5Y cell models [43], a neuroprotection role based on the upregulation of Nrf2 was reported. In the PC12 study of Cheng et al. [52], the Rb1 activated SIRT1 and inhibited TLR4/MYD88 protein expression. The activation of SIRT1 affected the deaceylation of NF-kB, which then lead to a reduction of TNFa, IL-2, and IL-6 inflammatory factor production. The parallel inhibition of the expression of TLR4 and MY88 protein expression in the penumbra converged with the effects of the SIRT1 activation to affect a type of molecular pincer action, reducing cellular inflammatory processes. Thus, the integrated activities of Nrf2/ARE and SIRT 1/TLR4 provide a mechanistic foundation by which Rb1 affects chemoprotection.

### 5.2. Neuronal Repair/Regeneration Role of Schwann Cells

Schwann cells, glial cells of the peripheral nervous system, have generated considerable interest for their potential application in neural tissue engineering. Schwann cells secrete bio-activation agents that enhance axonal outgrowth and migration. In fact, Schwann cells enhanced nerve cell regeneration in large nerve gap lesions [70,71]. Such observations have resulted in Schwann cells becoming an important addition to tissue engineering nerve grafting activities. However, the biological properties of Schwann cells typically diminish significantly during their preparation time, limiting their practical application in the construction of artificial nerve grafts. As a result of such issues, efforts have been made to upregulate key nerve repair properties of Schwann cells by electrical stimulation [72,73] and via drug treatment [74]. Exploring this issue further, Liang et al. [75] assessed the possible capacity of Rb1 and Rg1 to enhance Schwann cell reparative functions across a 10-fold concentration range. The authors justified their study on the basis that Rb1 and Rg1 promoted neurite outgrowth along with enhancing protection in various preconditioning studies, which are assessed in the present paper. In their study, Liang et al. [75] measured the effects of Rb1/Rg1 on Schwann cell number, intracellular cAMP, PKA activity, and B-NGF and brain-derived neurotrophic factor (BDNF) protein levels. In each case, there was a biphasic dose response for both Rb1 and Rg1. Likewise, the use of a PHA inhibitor (H89) blocked each stimulated endpoint. While these findings demonstrated that Rb1 and Rg1 enhance the proliferation of Schwann cells, there has not been a significant clinical follow up. While the reasons are not clear for this lack of application, it should be pointed out that there is much competition for technology and agents, which may affect subsequent research activities.

### 5.3. Heart: Cardiomyocytes

Ginsenoside Rb1 was shown to protect heart cells in a series of experimental studies [76,77,78]. These investigations involved the protection of cardiomyocytes from hydrogen peroxide-induced oxidative stress. The Rb1-induced protective effects were mediated via the suppression of JNK activation [77]. A second report displayed Rb1-induced protection against damage from ischemia-reperfusion (i.e., myocardial infarction/reperfusion) injury in diabetic rats via activation of the P13K/AKT pathway [78]. Rb1 also protected against myocardial ischemia injury via the enhanced expression of eNOS [76]. These initial findings lead to follow-up investigations concerning whether Rb1 could protect neonate rat cardiomyocytes from hypoxia/ischemia (HI) oxygen (10%) induced damage using either a preconditioning or co-exposure protocol [79,80,81]. The HI experimental study utilized a six-hour preconditioning period followed by a 12 h HI exposure period. Rb1, which was tested over a concentration range of 3–160 μM, demonstrated a biphasic concentration response with protection being optimized at 40 μM, dropping significantly off from 80–160 μM (Figure 17) [80]. An assessment of five MiRNA, at the optimal Rb1 concentration (i.e., 40 μM) revealed that three MiR (Mir-1, Mir 29a, and Mir 208) were markedly increased. However, Mir 21 and Mir 320 were notably decreased. The authors particularly focused on the responses of Mir 21 and Mir 208 based on their specificity for heart tissue. A follow-up investigation extended the initial findings to another stressor, OGD, without a conditioning protocol [81]. The concentration range was reduced to 4–64 μM. As in the previous experiment, the dose response was biphasic, with the optimal response being 32 μM. Of interest was that the OGD-induced damage affected an increase in its target gene, programmed cell death protein 4 (PDCD4), which was the downstream target protein of Mir 21. This increase in PDCD4 was markedly reduced by Rb1. The effect of the Rb1 was blocked by an Mir 21 inhibitor, clarifying a mechanistic foundation for the Rb1-induced protective effects.

### 5.4. Diabetes

Ginseng has long been used in traditional Chinese medicine to treat type 2 diabetes. Various studies confirm that ginseng and some of its principal constituents induce hypoglycemia and insulin sensitizing effects. These findings raised further scientific questions concerning whether ginseng and/or its specific components may affect underlying factors such as adipocyte differentiation, glucose uptake into adipocytes and muscle cells, and insulin sensitivity. Since Rb1 is one of the most common ginseng constituents in multiple plant sources, it was selected for experimental evaluation related to the above question.

### 5.5. Adipocyte Differentiation

In their research, Shang et al. [82,83] reported that Rb1 enhanced the differentiation of 3T3-L1 pre-adipocytes to adipocytes. During the enhanced differentiation process, the Rb1 also increased the expression of mRNA and proteins of PPARy2, C/EBPa, and GLUT4. The Rb1 treatment biphasically enhanced basal and insulin mediated glucose uptake by adipocytes (Figure 18) and C2C12 myotubles (Figure 19). The quantitative features of these two biphasic dose responses were consistent with the hormetic dose response.

The mechanisms underlying the adipocyte differentiation and enhanced glucose uptake by the 3T3-1 adipocytes and C2C12 myotubes have been addressed to some extent. Shang et al. [82] proposed that Rb1 may directly bind to PPARγ2, affecting the over expression of PPARγ2 and C/EBPα, enhancing adipocyte differentiation. In fact, PPARγ2 is expressed selectively in adipocytes and is essential for the differentiation process. In addition, Rb1 also enhances glucose transport in mature adipocytes by the insulin signaling pathway. These findings suggest that Rb1 may affect insulin signaling via multiple processes.

### 5.6. White to Brown Adipocyte Transformation

The above findings have stimulated further research using 3T3-L1 adipocytes, especially in the area of transforming white adipocyte tissue (WAT) into brown [84]. This is of public health interest, since the so-called browning of WAT has been related to reducing obesity and insulin resistance. Using the 3T3-L1 adipocyte cell model, the Rb1 enhanced basal glucose uptake and the browning response based on the induction of multiple biomarkers. This Rb1-induced browning process also displayed the quantitative features of the hormetic dose response. This process was blocked by a PPAPγ antagonist.

Of particular interest was a study by Hosseini et al. [85] concerning an hormetic dose response for Rb1 on GLU4 gene expression in C2C12 cells after 12 h (Figure 20). The C2C12 cells have the capacity to develop into skeletal and cardiac muscle and are widely used as an effective predictive model. The concentration range over which the stimulation occurred was broad, being over five orders of magnitude. However, the maximal increase was distributed over about a 100-fold (0.1–10 μM) concentration range.

### 5.7. Wound Healing

Another aspect of the diabetes–ginseng interaction is that a local administration of saponin extract significantly enhanced wound healing in diabetic and aging rats [20]. In follow-up clinical trials, oral ginseng enhanced the repair of intractable skin ulcers in patients with diabetes mellitus [19]. Mechanistic follow-up studies have shown that the ginseng treatment enhances would healing via the stimulation of fibronectin synthesis in a hormetic dose response manner through changes in the TGF-β receptor in fibroblasts [19]. More recent experimental studies have shown the Rb1 biphasically enhances human dermal fibroblast proliferation and collagen synthesis (Figure 21) [86]. The authors suggested that their findings may be clinically applicable to wound-dressing strategies, suggesting the possibility impregnating ginseng mixtures into bandages. In fact, ginseng extracts have been reported to protect skin in mouse models from acute UVB irradiation [87] as well as to significantly enhance healing after laser burn injury and excisional wounding [88,89,90]. Multiple patents are listed in the Web of Science for products containing ginseng for use in wound healing. However, it does not appear that these developments have taken advantage of knowledge of the hormetic dose response.

### 5.8. Hair Growth

In 2020, Calabrese [91] provided detailed assessment of the occurrence of stimulating hair growth within experimental and clinical frameworks for almost 200 agents, including ginseng. The initial publication showing that ginseng could enhance hair growth was reported by Kubo et al. [92] with the ddy mouse strain, using the findings to obtain a patent in 1986. A spate of hormetic effects of ginseng constituents, including ginsenoside Rb, some its constituents such as Rb1, its metabolite F2, and Rb2 in the growth of hair was reported only within the past few years, following the development of several in vitro models such as human keratinocyte cells, mouse vibrissae hair follicles, and other cellular models (e.g., [93,94,95,96]). These hormetic studies with ginseng involved experiments with human hair dermal papilla cells (HHDPs), human keratinocyte cells (HaCaTs) (Figure 22), and other experimental models.

Amongst these findings were observations that hair growth was enhanced in a hormetic fashion; likewise, the Rb treatment prevented the occurrence of hair growth inhibition by dihydrotestosterone (DHT) within a preconditioning experimental framework (Figure 23). Whole animals studies using young adult C57BL/6 male mice extended these hormetic dose responses. These whole animal studies used an oral dosing protocol, making the study of particular relevance to humans, since the doses used are commonly ingested by people. The findings of mouse in vivo studies were consistent with the clinical findings of Kim et al. [97], who reported that orally administered Korean Red Ginseng enhanced hair density and thickness in alopecia patients with a dosage of 3000 mg/day for 24 weeks.

### 5.9. Ginseng-Rd

There has been growing interest in the assessment of the principal constituents of ginseng. This section will address the ginseng constituent Rd and how some of its biomedical effects are mediated via hormesis. Interest in Rd originated with the findings of Yokozawa et al. [98,99,100], showing that Rd displayed antioxidant effects in kidney injury models of senescence-accelerated mice. These initial findings would stimulate research interest in Rd but with a particular focus on neuronal cellular systems starting with the work of Lopez et al. [101], showing that Rd decreased ROS formation in cultured astrocytes.

### 5.10. Stroke

The first report of a hormetic dose response for Rd was made by Ye et al. [102] using PC 12 cells (Figure 24). In this study, the Rd was administered at the same time as the oxidant stressor, hydrogen peroxide. The Rd treatment displayed hormetic concentration responses for LDH release and cell survival. Low Rd concentrations prevented a complex set of mitochondrial damages by hydrogen peroxide. Linked to the induced protective responses were increases in antioxidant enzymatic activities of SOD and GPx. Several years later, the same researchers extended their hormetic research findings with Rd to in vivo stroke outcomes with an aged male mouse model. Using 16–18-month-old male C57BL/6 mice, the Rd protected against stroke-induced damage within pre- and post-conditioning protocols [103]. The preconditioning protocol utilized five doses (i.e., ranging from 0.1 to 200 mg/kg) (Figure 25), whereas the post-conditioning protocol used only the optimal dose of the preconditioning protocol but applied it at four different times after the induced stroke. In this study, the Rd was highly effective in reducing stroke damage both before (preconditioning) and after induction (post-conditioning). Follow-up mechanistic experiments by Ye et al. [103] confirmed that Rd mediated protection by preventing mitochondrial damage in the cortex and striatum by the upregulation of antioxidant enzyme activities linked to the enhancement of mitochondrial stability.

### 5.11. Other Neuroprotective Studies

Since the initial hormetic studies of Ye and colleagues, there has been a progressive stream of investigations with Rd on a spectrum of neuronal related experimental systems. Follow up studies with Rd revealed hormetic dose responses with (1) the embryonic mouse (Embryonic day—4.5) for neurosphere development [104] (Figure 26), (2) in SH-SY5Y neuronal cells in a preconditioning protocol (2 h) preventing MPP-induced toxicity (Figure 27) [105], (3) in HT22 mouse-derived hippocampal cells that prevented glutamate toxicity in a preconditioning protocol (2 h) (Figure 28) [106] and (4) in an in vivo preconditioning protocol with Sprague–Dawley rat spinal cord tissue preventing ischemia-induced injury (Figure 29) [107]. This complex array of experimental studies demonstrating Rd-induced hormetic dose responses revealed that the protective concentration range was similar, optimizing in the 1–10 μM range. Hormetic pathway analysis revealed that the protection could be blocked by inhibiting the P13K/AKT pathway [105].

Multiple research teams emphasized that RD is highly lipophilic and readily passes across the blood–brain barrier. It also has a long plasma half-life, being about 50–60 h in humans, and nearly 20 h in rats [108]. These factors are likely to encourage further research and possible clinical application considerations.

### 5.12. Heart

While the above research focused on the neuroprotective potential of Rd, Wang et al. [109] reported that Rd attenuated myocardial ischemia/reperfusion injury in both in vitro and in vivo studies. In in vitro experimentation, neonatal rat cardiomyocytes from two-day-old Sprague–Dawley rats were assessed in a preconditioning protocol to determine if Rd could protect against ischemia/reperfusion injury. The four concentration-based (0.1–50 μM) study revealed a hormetic concentration response with significant protection reported over the 0.1 to 10.0 μM range based on cell viability and LDH leakage (Figure 30). Extensive follow-up experiments demonstrated that Rd mediated the protection via enhancing mitochondrial membrane potential and preventing mitochondrial-mediated apoptosis. In a complementary in vivo study, the single dose preconditioning protocol reduced the cardiac damage by nearly 50%. The Rd exposure was via an IP administration that would be equivalent to 3500 mg per 70 kg person.

### 5.13. Kidney

In 2018, Lee et al. [110] assessed the capacity of Rb1, Rb2, Rc, Rd, Rg1, and R3 to prevent FK506 toxicity to LLC-PK cells, which is a porcine kidney epithelial cell model. This was undertaken since FK506 is highly nephrotoxic and widely used by humans to prevent organ rejection in transplantation procedures. Of interest to the present section is that Rb2, Rc, and Rd displayed hormetic dose responses in a preconditioning (2 h) protocol (Figure 31) [110]. The ginseng constituents were effective in preventing kidney damage by blocking intoxication pathways such as p38, KIM-1, and caspase 3.

### 5.14. Polyacetylenes

#### Introduction

Aliphatic C17-polyacetylenes of the falcarinol-type are commonly found in carrots and related vegetables such as parsely, celery, parsnip, and fennel. They are also present in the lipophilic fraction of ginseng. Such polyacetylenes may display anti-inflammatory activities [111]. Of particular relevance is that hormetic dose responses have been reported in the areas of neuroprotection [74,112] and tumor cell biology (Figure 32) [112] with several polyacetylenes [e.g., panaxydol (PND) and panaxyol (PNN)].

### 5.15. Alzheimer’s Disease

Nie et al. [112] reported that the pretreatment of primary cultured Sprague–Dawley rat embryonic (E-16) cortical neurons with PND or PNN prior (24 h) to beta-amyloid (25–35) exposure significantly enhanced cell survival as measured by the MTT assay (Figure 33). This protection was dose-related and displayed an hormetic dose response. Similar protection also occurred with PND and PNN when given at the same time as the AB 25–35 treatment. The protective effects were related to a suppression of apoptosis that was mediated by increases in the BCL-2/Bax ratio and enhancement of mitochondrial membrane stability. PND and PNN also reversed AB 25–35 induced calcium influx and intracellular free radical generation. The authors suggested that antioxidants with good blood–brain barrier permeability are needed for use in the prevention of neurodegenerative diseases such as AD. They noted that since PND and PNN are lipophilic and found in many food plants, that ingestion of such plants may offer benefit against such neurodegenerative diseases.

### 5.16. Nerve Regeneration

As a result of the neuroprotective findings of PND/PNN, He et al. [74] assessed whether PND might enhance the capacity of Schwann cells for growth and healing following damage. Using Schwann cells from the sciatic nerve of newborn (1–3-day-old Sprague–Dawley) rats, it was determined that PND enhanced the expression and secretion of nerve growth factor (NGF) and brain-derived neurotrophic factor (BDNF) displaying hormetic dose responses (Figure 34). At the peak level of NGF and BDNF secretion, the Schwann cells also enhanced the synthesis of actin, which is an important component of the cytoskeletin. Similar studies also revealed that the optimal concentration for NGF/BDNF secretion also improved mitochrondrial transmembrane potential. The authors concluded that PND had the potential to enhance repair in neurons by its actions with Schwann cells.

### 5.17. Rg3

#### Pain/Tobacco Toxicity/Diabetes/Vero Cells

A limited number of hormetic dose responses have been reported for Rg3. These include the effects of R3 (20(s)-Rg3 isomer) on plantar incisional pain in an adult Sprague–Dawley rat model. Inflammatory cytokines levels closely correlated with the pain response (Figure 35) [113], protection against cigarette smoke extract-induced cell injury in a preconditioning protocol [114], the biphasic enhancement of glucose-stimulated insulin secretion, and AMPK activities by both the 20R and 20S isomers [115] and protection against glutamate-mediated neuronal cell death to HT22 cells, a mouse hippocampal cell line (Figure 36) [106], and protected against ginsenoside induced cell toxicity in Sprague–Dawley fetal cortical cells. With respect to the biphasic dose response for plantar incision pain, the authors suggested that this could make it more challenging for use in a therapeutic application, noting further the need to document the biphasic dose response in humans. In the present study, the mechanism of pain reduction was not related to opioid, nicotinic, or muscarinic acetylcholine receptors based on experiments with antagonists. However, alpha-2 adrenergic receptors were involved with the pain modulation of Rg3. For example, yohimbine, an alpha 2-adrenergic receptor antagonist, blocked the analgesic effect of Rg3. Finally, 20(S)-Rg3 enhanced the cell proliferation of Vero cells in an hometic dose response manner (Figure 37) [116]. The experiment was designed to assess the capacity of 20(s)-Rg3 to inhibit the growth of herpes simplex viruses (types 1 or 2). In the presence of this evaluation, the hormetic effects on Vero cells were observed. A similar hormetic effect on Vero cells was reported earlier by Song et al. [117] for multiple ginsenosides but only using three concentrations rather than the six concentrations of the Wright and Altman [116] study.

## 6. Ginseng Constituent Re

### 6.1. Introduction

The ginsenoside Re has been reported to display hormetic dose responses, but to a more limited extent than Rg1, Rb1, and Rd. Amongst these hormetic publications were two papers using HUVEC to assess whether Re may reduce oxidative stress and its potential value in diminishing the risk of developing atherosclerosis [118,119]. The remaining three hormetic papers dealt with (1) the capacity of Re to promote nerve cell regeneration by enhancing the proliferation, differentiation, and migration of Schwann cells in the Sprague–Dawley rat model of sciatic nerve crush injury [120], (2) the capacity of Re to prevent toxicity induced by beta amyloid in SH-SY5Y cells [121], and (3) whether Re could block I/R toxicity using PC 12 cells [52].

### 6.2. Nerve Injury and Regeneration

Using as its rationale the results of a series of a studies with Re on gastric muosal lesions [122], renal ischemia–reperfusion injury [120], and several other only tangentially related protective activities, a sciatic nerve study was undertaken using adult male Sprague–Dawley rats. In this study, six doses served as pretreatment ranging from 0.5 to 3.0 mg/kg [120]. Only the data for the 2.0 mg/kg treatment group were shown, being referred to as the optimal pain-relieving dose. However, the entire dose range response was provided for changes in PCNA expression. For this endpoint, the 1.0–2.0 mg/kg dose range response was significantly increased with the 2.0 mg/kg group showing the largest gain (133% versus 100% control) (Figure 38). This study demonstrated that Re enhanced sciatic nerve repair via multiple hormetic processes.

### 6.3. Alzheimer’s Disease

With respect to the beta-amyloid induced toxicity in SH-SY5Y cells, the Re prevented cytotoxicity (Figure 39) and apoptosis by inhibiting the beta-amyloid-activated mitochondrial apoptosis pathway (as inferred by its effects on mitochondrial functions, BCL-2/Bax ratio, cytochrome c release, and caspase 3/9 activities [121]. Re treatment also blocked JNK activation while upregulating NrF2. Furthermore, blocking NrF2 by interfering RNA abolished the protective effects of Re.

### 6.4. Atherosclerosis

Two papers concerning Re and HUVEC displayed hormetic doses responses relating to reducing risks of developing atherosclerosis. However, both studies did so within the context of addressing complementary research questions. The Huang et al. [118] study increased cell proliferation in both direct stimulation and preconditioning experimental protocols. In the case of the preconditioning protocol, the range of conditioning doses displayed the low dose stimulation but was not evaluated at higher doses to sufficiently assess possible inhibitory effects.

The Yang et al. [119] study did not address the issue of a direct low dose stimulation but demonstrated an hormetic dose response in a preconditioning protocol using a broader dose range which had a 3.5-fold higher maximum dose tested than in the Huang et al. [118] study. This study also used Ox-LDL as the stressor rather than hydrogen peroxide. Nonetheless, both studies used the same cell model, offering complementary perspectives. Finally, the study by Cheng et al. [52] was unique as it assessed five ginsenosides using the PC 12 cell model with pre- and post-conditioning protocols with hormetic dose responses being shown for each ginsenoside (Figure 40). This study is discussed in greater detail in the Rg1 and Rb1 sections of this paper.

### 6.5. Gintonin

#### Introduction

Gintonin, a component of ginseng, is an exogenous ligand for G-protein-coupled lysophosphatidic acid (LPA) receptors. Gintonin is a non-saponin and non-acidic carbohydrate polymer. The gintonin-enriched extract (GEF) is comprised of a large proportion of gintonin, linoleic acid, phosphatidic acid (Pa), and other bio-reactive lysophospholipids. The unique feature of gintonin is that it uses G-protein-coupled LPA receptor signaling pathways by which gintonin induces a broad spectrum of hormetic dose responses that affect (1) human dermal fibroblast cell proliferation [123,124], (2) human/rabbit corneal cell repair [125], (3) modulation of the mouse blood brain barrier [126], (4) angiogenesis and wound healing using HUVEC models [127], (5) glycogenolysis and astrocyte preconditioning [128], and (6) beta-amyloid-induced dysfunction in mouse models via cholinergic stimulation of ACH release [129].

### 6.6. Corneal Injury

Of particular interest was research that employed human corneal epithelium (HCE) cells to evaluate the capacity of gentonin to affect healing-related processes. Kim et al. [125] demonstrated hormetic dose responses for ERK1/2 phosphorylation for epithelial cell proliferation and migration. These effects were blocked by the LPA1/3 receptor antagonist Ki16425, phospholipase C (PLC) inhibitor U73122, inositol 1,4,5-triphosphate receptor antagonist 2 APB and intracellular Ca2+ chelator BAPTA-AM. Of significance was the potential capacity to enhance corneal wound healing when applied to in vivo rabbit models with induced corneal damage. Using an ocular dose of 200 μg gentonin, the healing process was significantly enhanced several fold, depending on the endpoint measured.

### 6.7. Alzheimer’s Disease

In vitro and in vivo studies assessed the area of scopolamine and beta amyloid-protein mouse models of AD [129]. Since the LPA receptor has an important role in learning and memory in aged animals, gintonin was assessed for its capacity to block effects of beta-amyloid on brain functioning. Research with male ICR mice demonstrated that gentonin stimulates the release of ACH from cells expressing an endogenous LPA receptor in a hormetic dose–response manner. These findings complemented studies showing that the oral administration of gintonin reversed scopolamine-induced memory dysfunction, blocked beta-amyloid-induced reduction of ACH (Figure 41) and choline acetyltransferase (CHAT), and diminished ACHE activity in the mouse hippocampus. Gintonin also blocked amyloid plaque-induced changes in ACH, CHAT, and ACHE in a transgenic AD model. These findings for gintonin are consistent with FDA-approved drugs for the treatment of AD with respect to their hormetic effects at the level of operational mechanisms [130].

### 6.8. Astrocyte Glycogenolysis and Preconditioning

Astrocytes are brain cells that store glycogen. These cells also express lysophosphatidic acid (LPA) receptors. Astrocytes provide energy for neurons via astrocytic glycogenolysis under both physiological and pathological frameworks. Ginseng has been linked to this astrocyte–neuron interaction, since gintonin is an exogenous protein-coupled LPA receptor ligand. As a result of this biological framework, Choi et al. [128] assessed the capacity of gintonin to affect astrocyte glycogenolysis, ATP production, glutamate uptake, and cell viability under a range of biological conditions, including hypoxia and re-oxygenation stress states. The gintonin treatment increased astrocyte glycogenolysis via LPA receptors under both stress and non-stress-related conditions in an hormetic dose–response fashion. Thus, gintonin can mediate astrocyte energy release as well as protect and enhance the adaptive capacity of astrocytes under stress following hormetic dose response processes. These findings have potentially significant neuroprotective implications for public health and clinical application. Gintonin has also been shown to hormetically enhance the blood–brain barrier permeability and brain delivery via the use of LPA receptors, which is a process that may be of utility with respect to new formulations of ginseng mixtures but also combined with other potentially beneficial agents that have poor capacity to cross the blood–brain barrier [126].

### 6.9. Wound Healing

Several research initiatives have shown that gentonin has the potential to enhance wound healing endogenously as well as with skin (Figure 42) [118,123]. Both studies reflect the capacity of gintonin to mediate these processes via hormetic dose responses, which are linked to LPA receptors. These findings have the potential to provide insight for how ginseng mixtures have been able to successfully treat various types of skin wounds within animal model studies and humans.

## 7. Ginseng Mixtures

### 7.1. Introduction

Ginseng has been a studied in a dose–response fashion via the use of various types of mixtures and individual constituent agents, such Rg1, Rb1, Rb2, Rd, Re, and others. In this section, the occurrence of hormetic dose responses as induced by several types of ginseng mixtures is addressed. Such mixtures include extracts from Korean Red Ginseng (KRG), Ginseng Total Saponins (GTS), and other ginseng mixture extracts [132]. In general, ginseng saponins may be placed into one of three major groups based on chemical structures: protopanaxadiols, protopanaxatriols, and oleanolic acid saponins. Protopanaxadiols (e.g., Rb1, Rb2, Rc, Rd, Rg3) have sugar moieties attached to the 3-position of the dammarene-type triterpine. In contrast, protopanaxatriols (Re, Rf, Rg1, R2, Rh1) have the sugar moieties attached to the 6-position. The roots of ginseng also contain other compounds such as polysaccharides, peptides, polyacetylene alcohols (e.g., panaydol), fatty acids, and minerals.

A comparison of KRG with GTS and various other ginseng extracts reveals that the ginsenoside constituent content of KRG is consistent across studies, with the following approximate common constituent ratios: with Rb1 (33%), Rc (13%), Rb2 (11%), Re (8%), and Rg1 (8%). The remaining constituents (e.g., Rd, Rf, Rh, Rg2) are typically in the 4–5% range. In the total saponin mixture papers of Xia et al. [23] and Hu et al. [22] from the same laboratory, the values were far different than those seen for KRG: Re (27%), Rg 2 (19%), Rg1 (9%), and Rb2 (7%). The remaining ginsenosides ranged from 1.5% to 5.5%. Another ginseng mixture (e.g., Panax notoginseng) provides markedly different proportions of chemical constituents [21]. Some authors purchase the extracts from the same company with the same ginseng extract (G115) with 4% total ginsenosides [13] but did not present the specific constituent values. In general, the marked diversity of ginseng constituent mixtures makes such studies unique experimental systems and difficult to directly compare with other ginseng products.

From an overall perspective, the ginseng mixture studies showing hormetic dose responses used 16 different biological models (Table 1) with eight being neuronal-behavorially related models. These mixture studies also included several pre- and post-conditioning experiments [22,23,24,25,26,133]. The quantitative features of the ginseng mixture hormetic dose response studies had a median maximum stimulation of 155.5% with the stimulatory dose range being 7-fold (Table 4).

It is not common for a total ginseng saponin mixture to be compared directly with the effects of specific constituents in the same study. However, in the report of Liao et al. [24], Rb1, Rg1 and GTS mixture (Figure 43) showed hormetic dose responses for spinal cord survival with embryonic Sprague–Dawley rats with both a direct stimulation and preconditioning protocols.

### 7.2. Kidney Toxicity

In the case of the KRG studies, hormetic dose responses were reported using five biology models (i.e., Dermal papilla cells, Park et al. [95]; A549 cell-human pulmonary epithelium cells, Song et al. [16,134]; HUVEC [15]; Murine osteoblastic MC3T3 cells [129,131]; LLC-PKI cells [133]) with a focus on possible clinical translation for preventing kidney toxicity during organ transplant procedures [133] and preventing osteoporosis as a result of long-term glucocorticoid treatment [131]. In the case of kidney toxicity, it is well known that the drug FK506 plays a critical role in assuring that a patient will not reject a transplanted organ. However, about 60% of patients report kidney toxicity in this process due to the FK506. The clinical importance of the adverse response to FK506 stimulated consideration to whether various chemopreventive agents might mitigate such adverse effects. While the first efforts in this general area were published by Hisamura et al. [135,136] using green tea extracts and tea polyphenols, it was inspired by a series of papers a decade earlier by Yokozawa et al. [137,138,139,140] showing that green tea tannins reduced the progression and severity of renal failure in nephrectomized rats. Subsequently, Hisamura et al. [135] showed green tea extracts/tea polyphenols to protect against FK506-induced toxicity to LLC-PK1 cells that were derived from pig renal tubular epithelium.

These developments led to research showing that KRG could also be effective in preventing nephrotoxicity induced by gentamicin in rat renal tubular cells [141,142] and by chronic exposure to cyclosporine [143]. These findings led Lee et al. [133] to assess whether KRG would prevent the toxicity of FK506 when both were administered at the same time. Using the LLC-PKI-pig kidney epithelial cells as their model, a toxic dose of FK506 was selected to challenge cell survival, reducing it to 60%. However, the KRG reduced significantly the FK506 toxicity in a manner that followed a hormetic dose response (Figure 44), with the optimal response occurring at 50 μM. Mechanistic follow-up studies revealed that the KRG treatment prevented an FK506-induced increase in *p*-p38, KIM-1, cleaved caspase 3, and increases in TLR-4 mRNA, all of which comprised key components of the toxicity pathways. A subsequent supportive study by Zhang et al. [144] reported that Panax notoginseng saponin (PNS) protected against polymyxin E-induced nephrotoxicity in an in vivo mouse (IM administration) study (Figure 45). Follow-up experiments with mouse renal tubular cells showed that PNS enhanced cell viability and the expression of BCL-2 while restoring mitochondrial function.

### 7.3. Bone Toxicity

KRG reversed the toxic effect of dexamethasone on osteobastic MC3T3-E1 cells as measured by multiple toxicity endpoints [131]. Of particular interest was that the KRG prevented the capacity of dexamethasone to decrease the expression of osteogenic gene markers (e.g., ALP) in MC3T3-E1 cells in a hormetic fashion (Figure 46). These findings provide a foundation to further explore whether ginseng may be able to reduce glucocorticoid induced osteoporosis in an in vivo rodent model.

### 7.4. Learning and Memory

Considerable research was undertaken on the effects of ginseng extracts on a range of behavioral parameters in animal models and humans by VD Petkov in Bulgaria during the mid 1950s to mid 1980s. Some of these investigations suggested a biphasic dose dependency. For example, Petkov [145] reported that orally administered ginseng extract at 20 mg/kg for three days enhanced learning and memory in rats undergoing punishment-reinforced maze training. However, this improved learning response was not only absent at the dose of 100 mg/kg but decreased relative to the control group. This biphasic dose dependency was subsequently explored in male Wistar rats in a series of different endpoint specific behavior/learning experiments using six doses with a dose range from 3 to 300 mg/kg with these treatments administered over 10 consecutive days. Training began on the 10th day, one hour following the last ginseng treatment. Learning retention testing was administered from 24 h out to 14 days following the cessation of the training sessions. The authors reported the occurrence of biphasic dose responses for multiple endpoint tests (e.g., multiple learning/memory Shuttle box tests, two-way avoidance and punishment reinforcement testing, step-down passive avoidance tests, staircase maze testing). In each case, the dose response was biphasic, with the optimal dose ranging from 10 to 30 mg/kg, depending on the endpoint and experiment. These findings confirmed earlier reports emphasizing the biphasic nature of the ginseng extract effects. Petkov and Mosharrof [146] noted that multiple researchers assessing others drugs with cognition activating effects reported similar biphasic dose responses, creating a type of hormetic “therapeutic window”.

### 7.5. Brain Traumatic Injury

The IP administration of GTS has also been shown to reduce traumatic brain injury in male rats (Figure 47) [22,23] in a hormetic manner. These findings were consistent with the report that Panex ginseng extracts protect astrocytes from hydrogen peroxide-induced toxicity during preconditioning experiments [26].

### 7.6. Muscle Injury

Orally administered ginseng extract (G115 extract) (4% ginsenosides) was also protective of muscle from exercise-induced stress regardless of which type of muscle fiber was studied with Wistar rats [12,13]. The protection started at the lowest dose tested (3 mg/kg), continuing to 100 mg/kg. The ginseng was administered daily for a three-month period. At the end of the three-month period, the animals were given an acute exercise stress experience using a treadmill [i.e., 20 m/min (0% gradient) for 80 min/day]. One week prior to the intense exercise, the rats were acclimated on the treadmill at 15 min/day with the speed increasing from 10 to 20 m/min. In traditional oriental medicine, the prescription for ginseng has been reported to be between 10 and 100 mg/kg, thereby excluding the low (3 mg/kg) and high (500 mg/kg) doses used in the Voces et al. [13] studies. Therefore, the protection is seen at about one-third of the lowest recommended dose. Of further interest is that Yu et al. [147] showed that oral Rg1 supplementation at 0.1 mg/kg significantly protected against exercise-induced oxidative stress in skeletal muscle.

### 7.7. Liver Injury and Repair

In Traditional Chinese Medicine (TCM), Panax has been employed to treat patients for a spectrum of liver diseases. According to Hui et al. [148], Panax nontoginseng saponins (PNS) have been effective in the treatment of liver fibrosis. Likewise, PNS has been protective against triptolide [149] and ethanol [150]-induced liver damage. These findings lead Zhong et al. [151] to hypothesize that PNS may have protective effects against damage resulting from partial hepatectomy (PH). In their follow-up investigations, the effect of PNS on liver regeneration following PH was assessed in a mouse model. This in vivo experiment was preceded by an in vitro primary hepatocyte cell culture study that assessed the effects of a broad range of PNS concentrations (0.01–0.50 μg/mL) on cell viability/cell proliferation assays. In the in vitro experiments, the PNS induced an hormetic–biphasic concentration response over the 50-fold concentration range with the maximum stimulation being 122% compared to a control value of 100% (Figure 48). A follow-up experiment using a marker for cell proliferation also showed a biphasic concentration response with the same optimized concentration between the two experiments. Then, this optimized concentration was selected for a follow up in vivo experiment to test the capacity of PNS to enhance liver regeneration in the PH study protocol. The PNS treatment in the in vivo study resulted in an increase in the liver to body weight ratio, lower levels of AST/ALS and other biomarkers of less injury, and enhanced regeneration. Mechanistic evaluation linked these enhanced recovery activities with the upregulation of P13/AKT/mTOR and the AKT/Bad cell pathways. According to the authors, this study provides support for the use of PNS in clinical settings for patients experiencing PH and perhaps other types of liver damage.

## 8. Discussion

The present assessment reveals that hormetic dose responses have been commonly reported in the biomedical literature for ginseng constituents and mixtures. While this assessment addresses the principal ginseng mixtures and constituents hormetic effects have also been reported in less well-studied constituents (e.g., RH2 [52], Rc [110] and Rg 3 [106]). The range of endpoints assessed has been broad but dominated by the area of neuroprotection. The areas of greatest interest have been those dealing with cellular/animal models for AD [48] and PD [41,42] disease, followed by stroke [51,52,152,153], neuronal health/regeneration based on studies with spinal cord [24,107]/sciatic nerve [62] injuries, and pain [113]. It is remarkable that this broad range of endpoints from a similarly wide variety of biological models and experimental approaches (e.g., pre-post-conditioning, non-conditioning, different mediators of damage) display hormetic dose responses. These results have also been reported by numerous independent research groups. In most studies, there has been substantial mechanism-receptor/pathway findings to account for the protective effects.

There have been other targeted research areas for the chemoprotective properties of ginseng. For example, there has been substantial research concerning the capacity of Rb1 to prevent symptoms of diabetes within an hormetic dose–response context [82,83]. Given the widespread occurrence of diabetes within the human population, there has been interest in using ginseng products in the treatment of type 2 diabetes. However, the biphasic nature of the hormetic dose response poses important challenges to identify endpoint specific optimal doses, as well as how close the optimal dose may be to potentially harmful higher doses along with the issue of human inter-individual variability.

The ginseng hormetic findings have also shown potential application for the treatment of corneal lesions with chemoprotection findings for both in vitro and in vivo studies [125]. In a similar fashion, ginseng extract treatment reduced the occurrence of FK506-induced kidney toxicity in a spectrum of model experimental systems [133]. These findings have potential practical relevance, since a very high proportion of organ transplant patients that receive FK506 to prevent organ rejection develop kidney toxicity.

Regardless of the biological model, endpoint measured, constituent studied, or mixture type employed, the quantitative features of the dose responses have been similar and consistent with the vast literature for hometic dose responses in plants and animals in vitro and in vivo settings [4,153,154]. The maximum stimulations with ginseng have been generally within the 130–180% range, while the width of the stimulation for the above respective comparison groups is less than 50-fold (Table 4). While the principal focus has been has been on assessing the low-dose stimulation, the high dose inhibitory aspect of the hormetic dose response is often more limited, as some studies do not evaluate responses greater than the threshold or even achieve a threshold. Further, the assessment of mechanism for the low-dose stimulation was also more extensively pursued than for the higher dose responses. An extensive mechanistic assessment of hormetic dose responses [13] includes mechanisms for some hormetic dose responses in both the low and high dose zones for those interested.

While efforts have been made to assess the pharmacokinetic characteristics of ginsenosides, there is a general lack of comprehensive studies in the area. The ginsenosides have chemically complex structures and a diverse range of biotransformation pathways, making generalizations difficult. However, most of the ginsenosides have low gastrointestinal absorption, being less than 5–10% [155,156,157]. The half-life in humans for several of the ginsenosides falls within the 18–24 h. duration [53]. Furthermore, the concentrations of ginsenosides are typically about 10-fold higher in the plasma than in the brain, suggesting that there is limited permeability through the blood–brain barrier. These observations indicate that there is likely to be a considerable range of cellular concentrations of various ginsenoside constituents.

A review of the ginseng hormesis literature reveals that a strong majority have utilized in vitro methods with cell cultures. However, nearly 20 studies utilized in vivo animal model studies assessing responses in a diverse set of organs, including the brain (TBI, stroke, AD pathology, behavior), skeletal muscle, heart, peripheral nerves, pain, and various diabetic disease endpoints. While most of the in vivo studies employed an IP administration, several papers administered the ginsenoside via the oral route. Of interest is that three of these oral studies dealt with brain endpoints such as AD pathology [129], depression behavior and BNDF [28], and learning behavior [146]. This suggests that some ginsenoside constituents and their mixtures can have biological effects on neuronal endpoints at relatively low concentrations. The optimal exposure in the Petkov and Mosharrof [146] study was 10 mg/kg, which is a dose that is commonly used by humans.

It has been widely emphasized that ginsenosides and their metabolites have poor bioavailability and cannot reach the intended biological targets when administered orally [158]. However, there are a large number of studies demonstrating that orally administrated ginseng mixtures/constituents have biologically significant responses at doses that are in the low to moderate range, affecting a broad range of organs and endpoints. These responses are seen for multiple organs such as the liver [100,159], kidney [133], brain/behavior [53,160], skeletal muscle [12,13,161,162], and others in experimental systems. Furthermore, a long-term (90 day) oral consumption of ginseng extract markedly reduced the susceptibility of both young (6-month-old) and intermediate-aged (18-month-old) rats to acute ischemia reperfusion injury following a preconditioning exposure [163]. Therefore, these examples indicate that there is convincing evidence that the statement by Yu et al. [158] relating to oral ginseng not reaching biological targets to have many exceptions and can not therefore be seen as reflecting a reliable generalization.

Despite the rather extensive number of reports in the biomedical literature that demonstrate that hormetic dose responses are common for ginseng effects, this is the first paper to provide an integrated synthesis of this topic. The implications of the present findings have the potential to be significant, since a very broad array of cell types and endpoints have been reported to display hormetic dose responses, often with variable optimal dose ranges. Humans ingesting ginseng mixtures in the 500 to several thousand mg/day range are likely to be common. Such exposures are expected to have the potential to induce a spectrum of biological effects, with some likely in hormetic optimal dose zone. However, despite the substantial recognition that ginseng dose responses are very commonly hormetic, the clarification of what the effects will be on human populations remain to be resolved. Furthermore, the large number of hormetic dose response observations in the experimental studies reported here should also encourage researchers to carefully consider their study design strategies with respect to the number and spacing of doses.

## Figures and Tables

**Figure 1 molecules-25-02719-f001:**
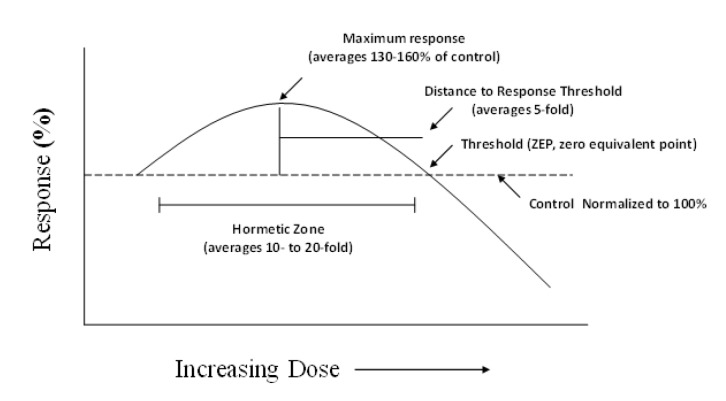
Dose–response curve representing the quantitative features of hormesis (Source: [7]).

**Figure 2 molecules-25-02719-f002:**
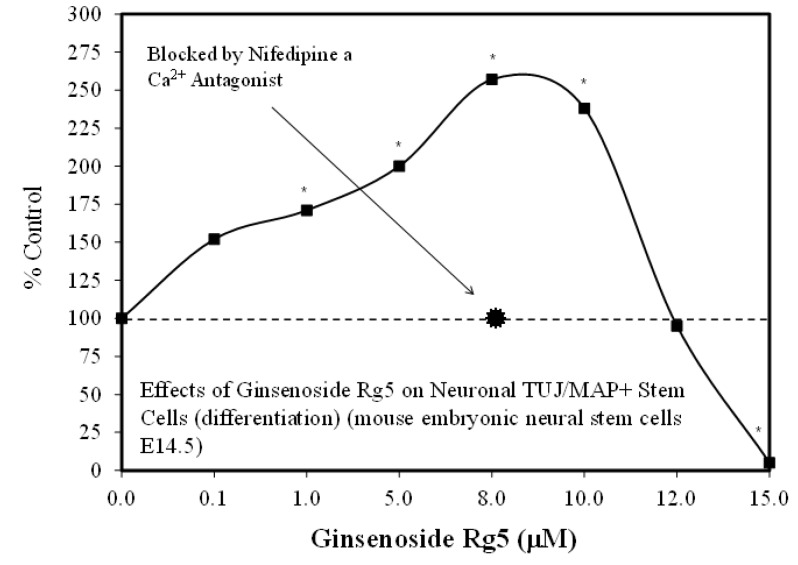
Effects of ginsenoside Rg5 on neuronal TUJ/MAP+ stem cells (differentiation) (mouse embryonic neural stem cells E14.5) (source: [33]). * = statistically significant

**Figure 3 molecules-25-02719-f003:**
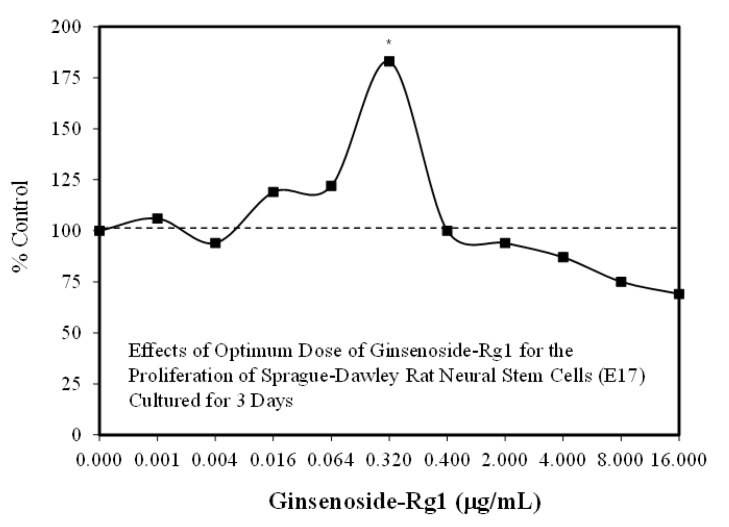
Effects of optimum dose of ginsenoside-Rg1 for the proliferation of Sprague–Dawley rat neural stem cells (E17) cultured for 3 days (source: [35]). * = statistically significant.

**Figure 4 molecules-25-02719-f004:**
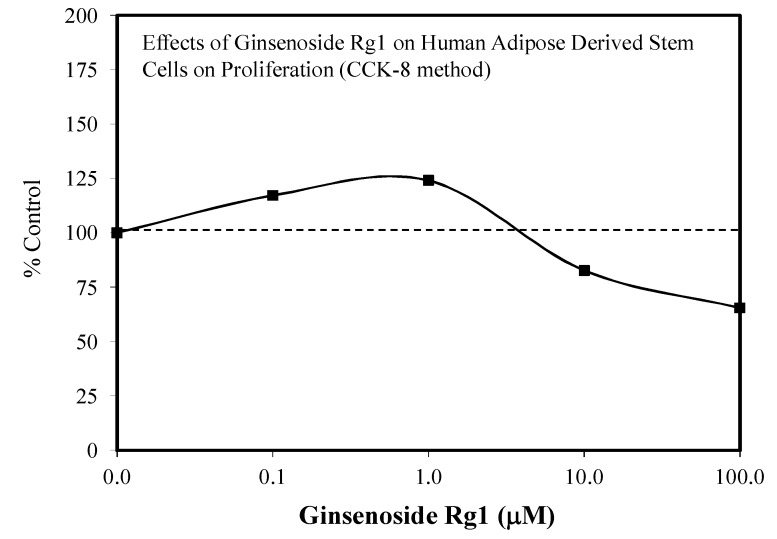
Effects of ginsenoside Rg1 on human adipose-derived stem cells on proliferation (CCK-8 method) (source: [34]).

**Figure 5 molecules-25-02719-f005:**
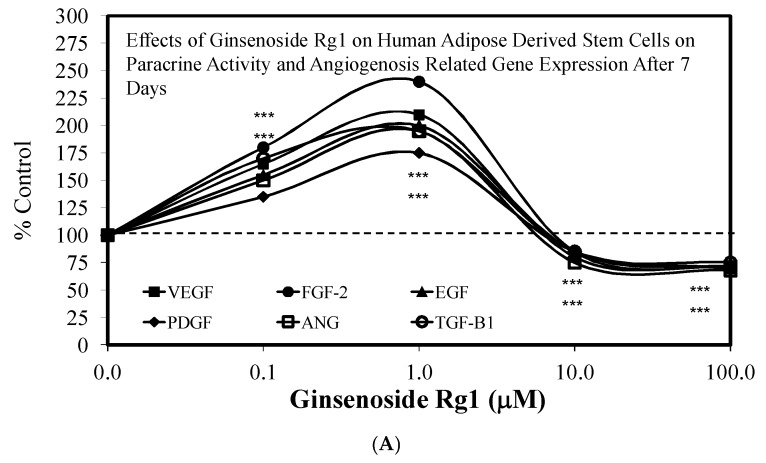
(**A**) Effects of ginsenoside Rg1 on human adipose-derived stem cells on paracrine activity and angiogenosis-related gene expression after 7 days (source: [34]). (**B**) Effects of ginsenoside Rg1 on human adipose-derived stem cells on paracrine activity and angiogenosis-related gene expression after 7 days (source: [34]). Each * represent statistical significance for a data point.

**Figure 6 molecules-25-02719-f006:**
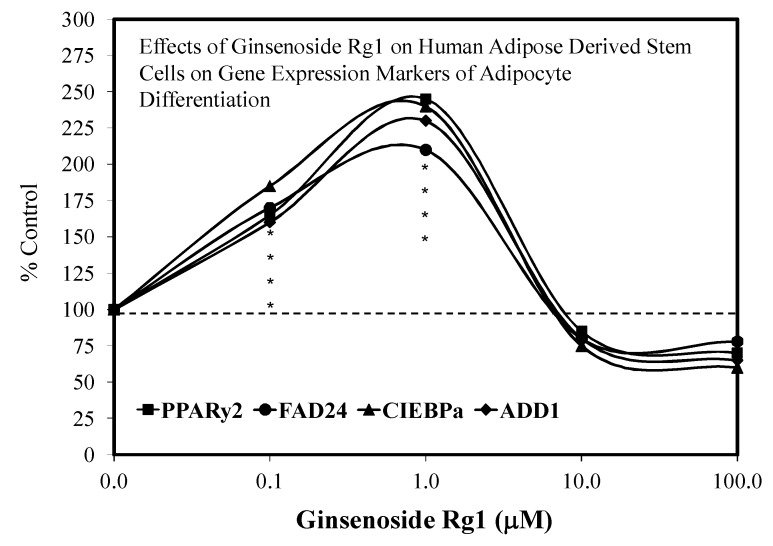
Effects of ginsenoside Rg1 on human adipose-derived stem cells on gene expression markers of adipocyte differentiation (source: [34]). Each * represent statistical significance for a data point.

**Figure 7 molecules-25-02719-f007:**
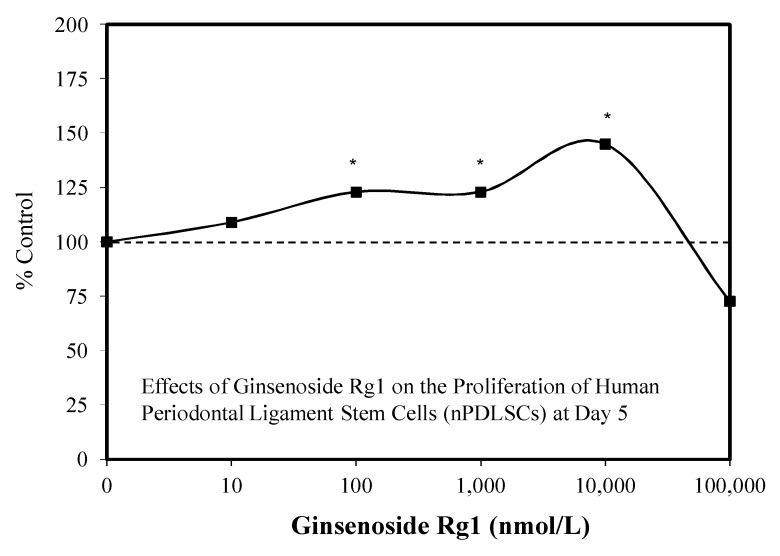
Effects of ginsenoside Rg1 on the proliferation of human periodontal ligament stem cells (nPDLSCs) at Day 5 (Source: [37]). Each * represent statistical significance for a data point.

**Figure 8 molecules-25-02719-f008:**
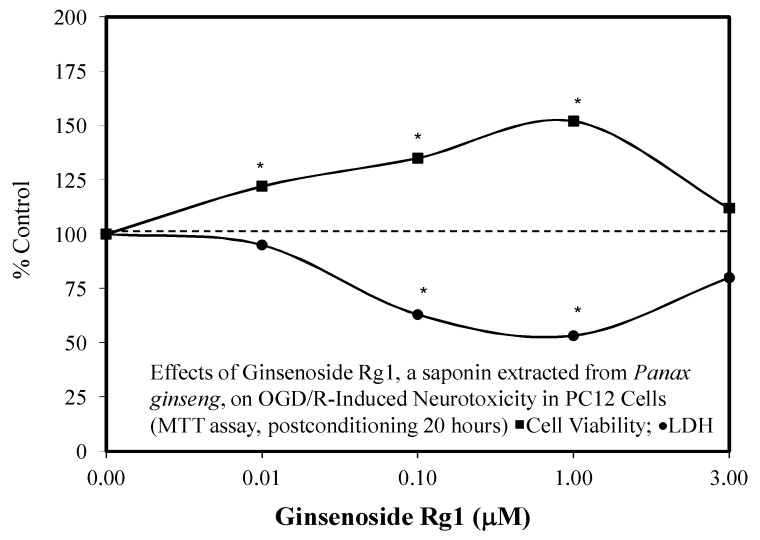
Effects of ginsenoside Rg1, a saponin extracted from *Panax ginseng*, on oxygen glucose deprivation(OGD)/reperfusion (R)-induced neurotoxicity in PC12 cells (MTT assay, postconditioning 20 h) (Source: [51]). Each * represent statistical significance for a data point.

**Figure 9 molecules-25-02719-f009:**
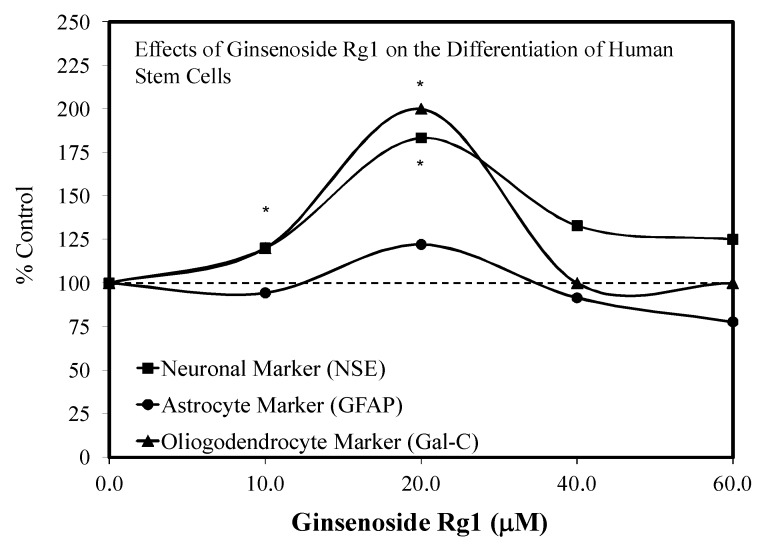
Effects of ginsenoside Rg1 on the differentiation of human stem cells (source: [39]). Each * represent statistical significance for a data point.

**Figure 10 molecules-25-02719-f010:**
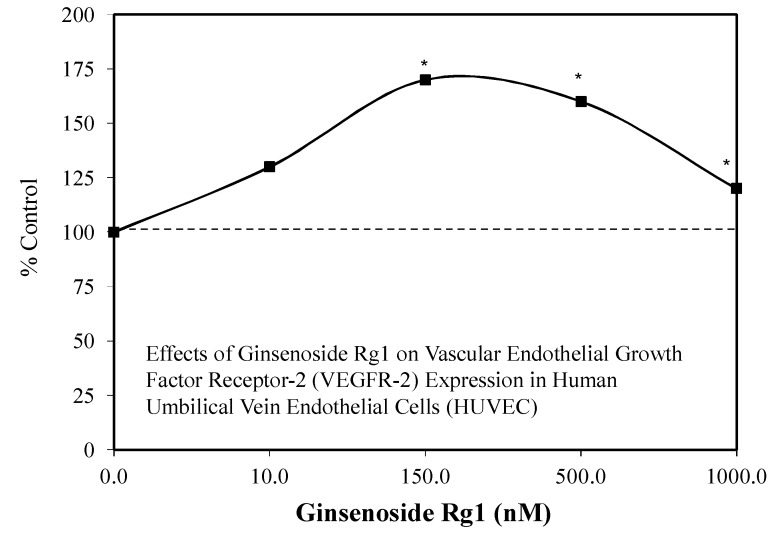
Effects of ginsenoside Rg1 on vascular endothelial growth factor receptor-2 (VEGFR-2) expression in human umbilical vein endothelial cells (HUVEC) (Source: [56]). Each * represent statistical significance for a data point.

**Figure 11 molecules-25-02719-f011:**
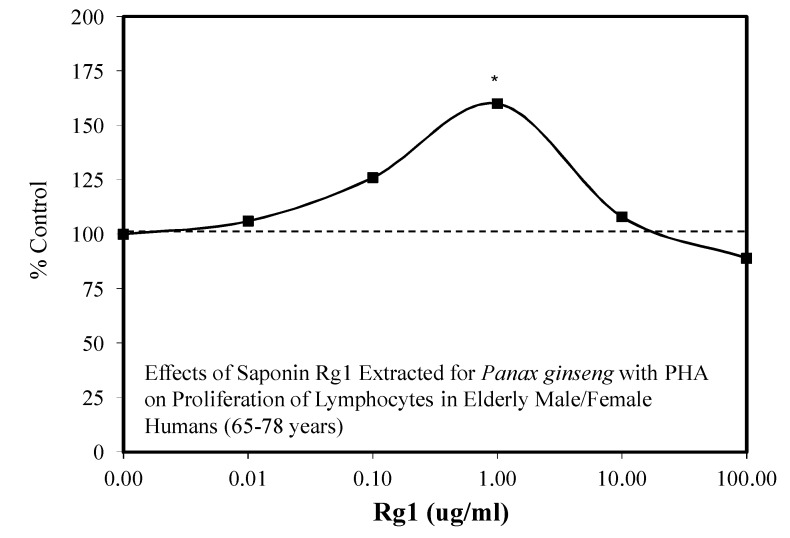
Effects of saponin Rg1 Extracted for *Panax ginseng* with phytohemagglutinin (PHA) on the proliferation of lymphocytes in elderly male/female humans (65–78 years) (source: [65]). Each * represent statistical significance for a data point.

**Figure 12 molecules-25-02719-f012:**
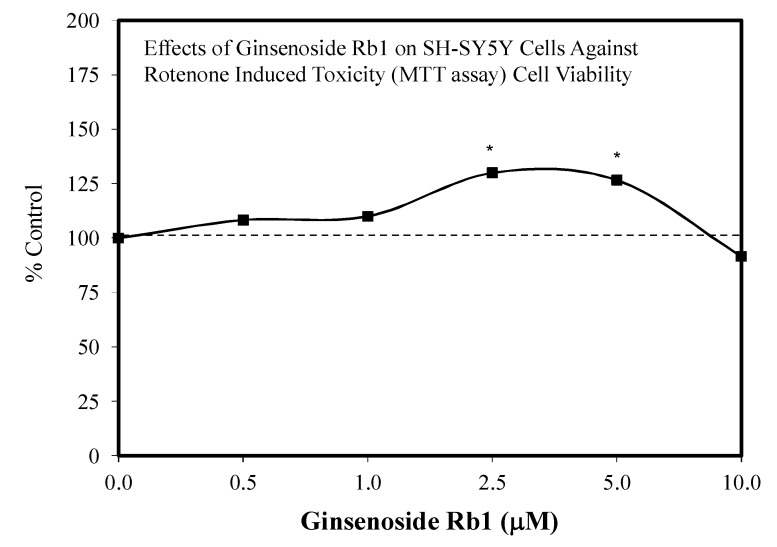
Effects of ginsenoside Rb1 on SH-SY5Y cells against rotenone-induced toxicity (MTT assay) cell viability (source: [43]). Each * represent statistical significance for a data point.

**Figure 13 molecules-25-02719-f013:**
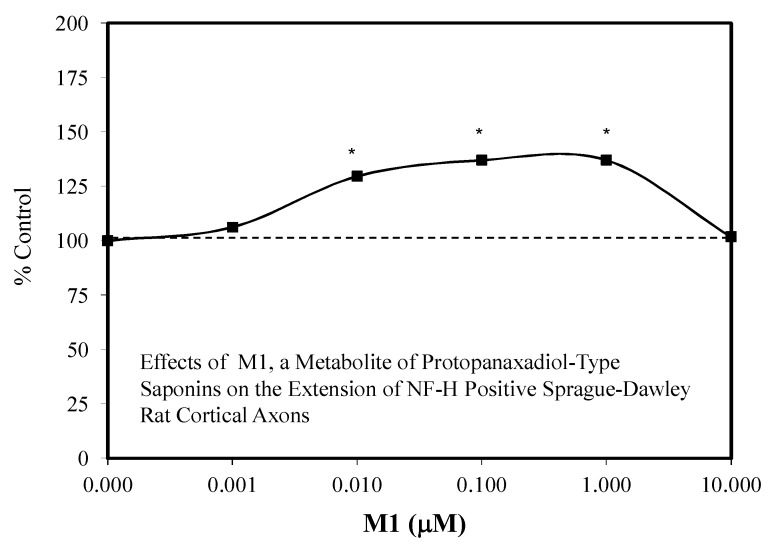
Effects of M1, a metabolite of protopanaxadiol-type saponins on the extension of NF-H positive Sprague–Dawley rat cortical axons (source: [67]). Each * represent statistical significance for a data point.

**Figure 14 molecules-25-02719-f014:**
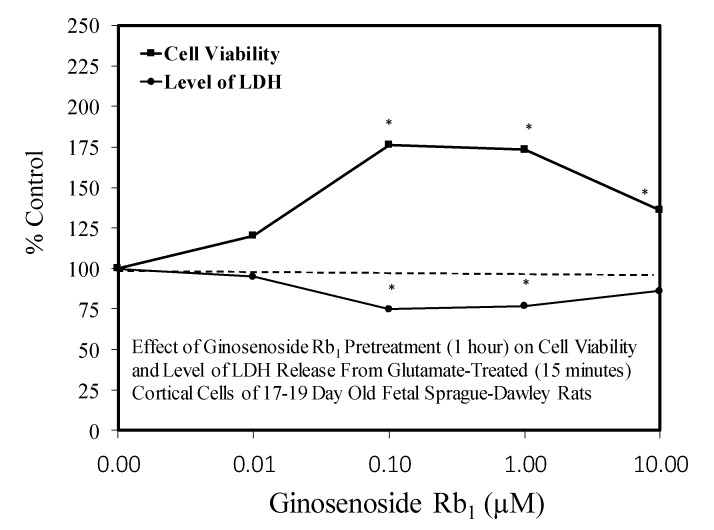
Effect of ginosenoside Rb_1_ pretreatment (1 h) on cell viability and level of LDH release from glutamate-treated (15 min cortical cells of 17–19-day-old fetal Sprague–Dawley rats (source: [68]). Each * represent statistical significance for a data point.

**Figure 15 molecules-25-02719-f015:**
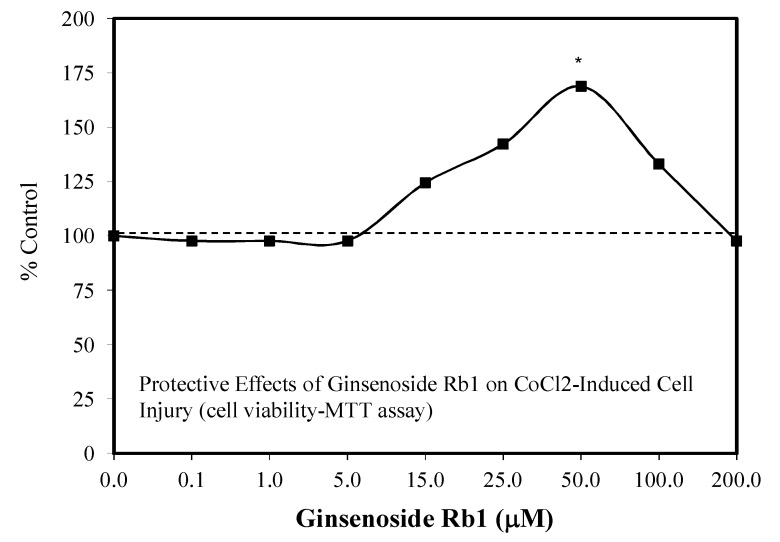
Protective effects of ginsenoside Rb1 on CoCl2-induced cell injury (cell viability: MTT assay) (source: [52]). Each * represent statistical significance for a data point.

**Figure 16 molecules-25-02719-f016:**
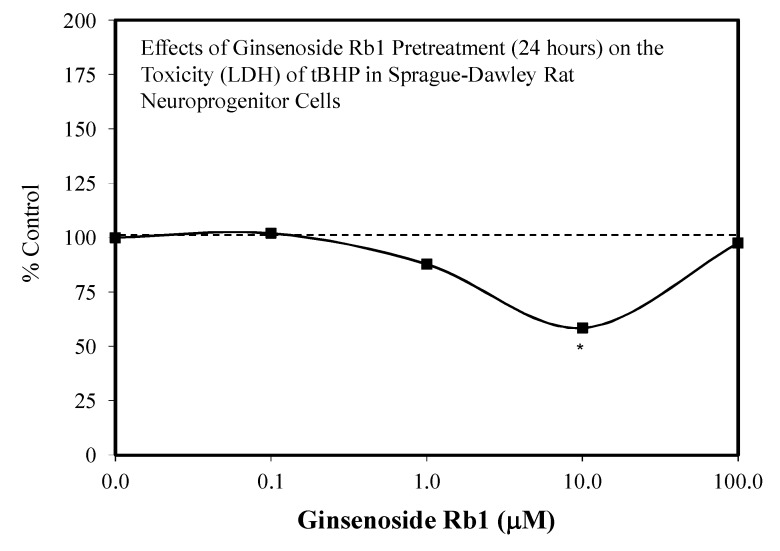
Effects of ginsenoside Rb1 pretreatment (24 h) on the toxicity (LDH) of tBHP in Sprague–Dawley rat neuroprogenitor cells (source: [69]). Each * represent statistical significance for a data point.

**Figure 17 molecules-25-02719-f017:**
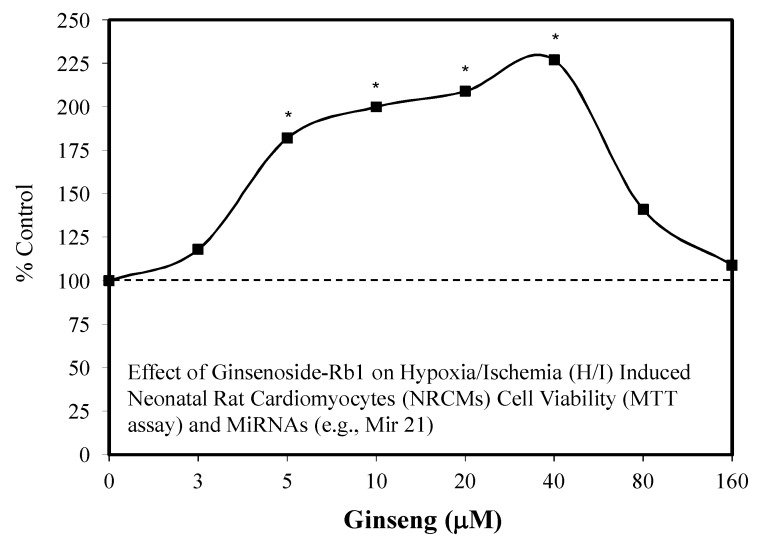
Effect of ginsenoside-Rb1 on hypoxia/ischemia (H/I)-induced neonatal rat cardiomyocytes (NRCMs) cell viability (MTT assay) and MiRNAs (e.g., Mir 21) (Source: [80]). Each * represent statistical significance for a data point.

**Figure 18 molecules-25-02719-f018:**
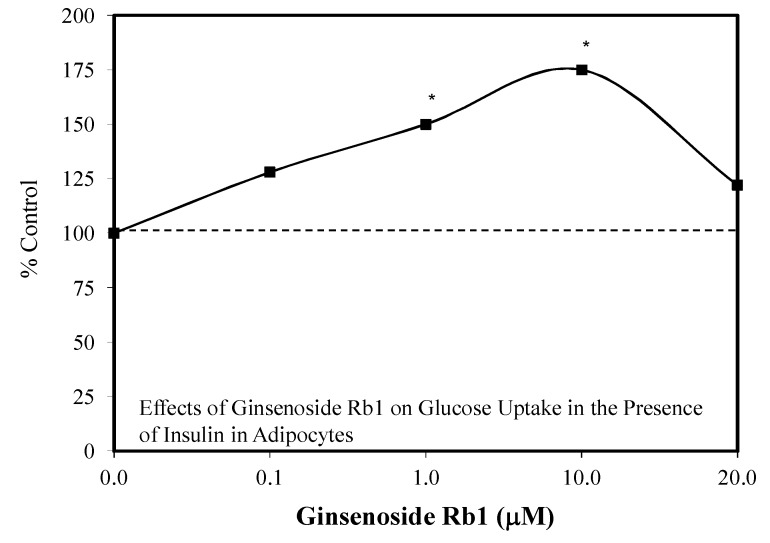
Effects of ginsenoside Rb1 on glucose uptake in the presence of insulin in adipocytes (source: [82]). Each * represent statistical significance for a data point.

**Figure 19 molecules-25-02719-f019:**
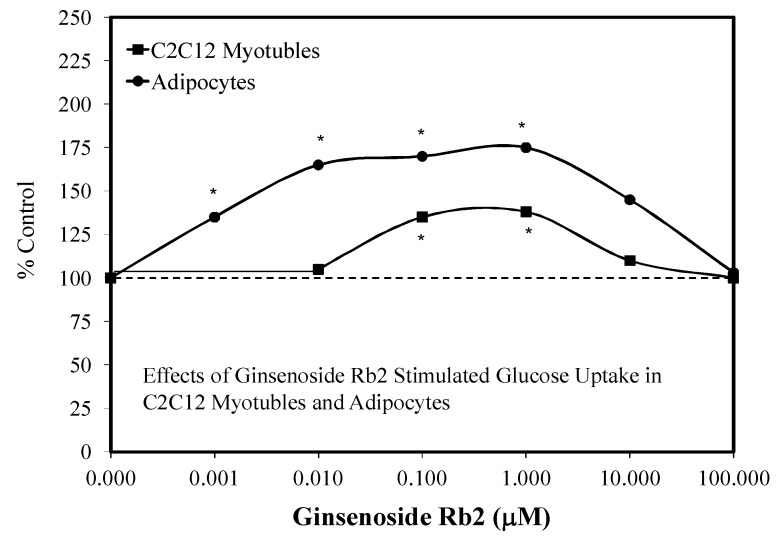
Effects of ginsenoside Rb2 stimulated glucose uptake in C2C12 myotubles and adipocytes (source: [83]). Each * represent statistical significance for a data point.

**Figure 20 molecules-25-02719-f020:**
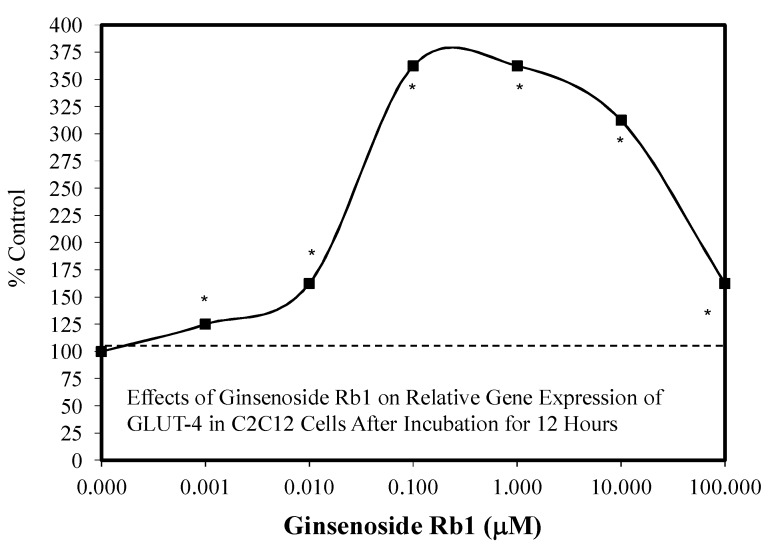
Effects of ginsenoside Rb1 on relative gene expression of GLUT-4 in C2C12 cells after incubation for 12 hours (source: [85]). Each * represent statistical significance for a data point.

**Figure 21 molecules-25-02719-f021:**
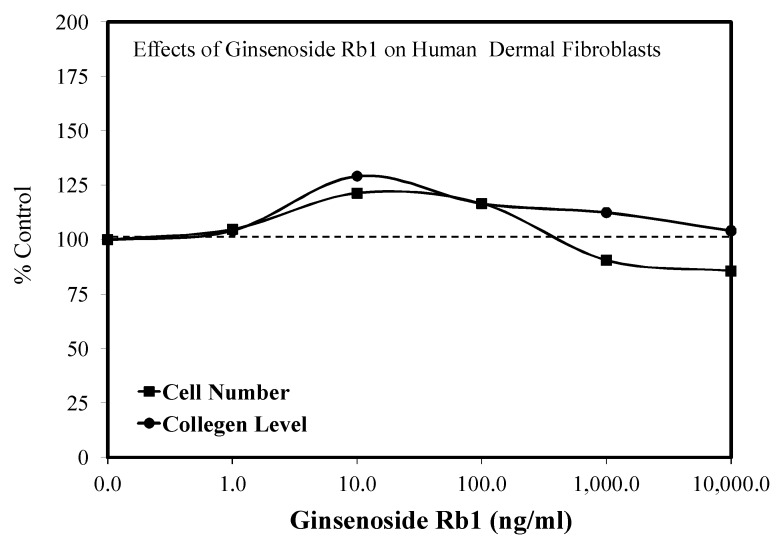
Effects of ginsenoside Rb1 on human dermal fibroblasts (source: [86]).

**Figure 22 molecules-25-02719-f022:**
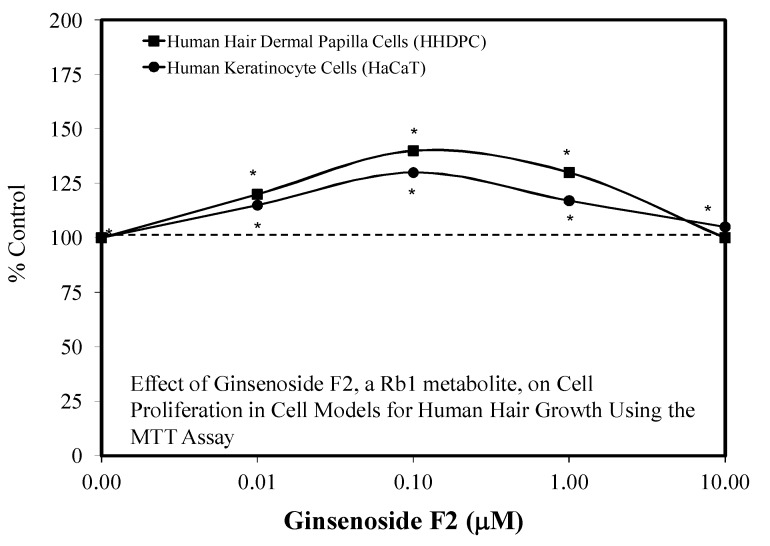
Effect of ginsenoside F2, a Rb1 metabolite, on cell proliferation in cell models for human hair growth using the MTT assay (Source: [93]). Each * represent statistical significance for a data point.

**Figure 23 molecules-25-02719-f023:**
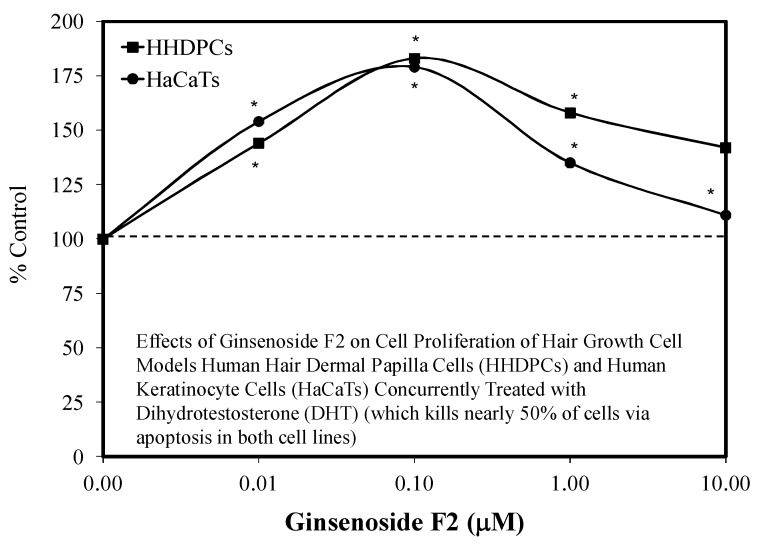
Effects of ginsenoside F2 on cell proliferation of hair growth cell models human hair dermal papilla cells (HHDPCs) and human keratinocyte cells (HaCaTs) concurrently treated with dihydrotestosterone (DHT) (which kills nearly 50% of cells via apoptosis in both cell lines) (Source: [94]). Each * represent statistical significance for a data point.

**Figure 24 molecules-25-02719-f024:**
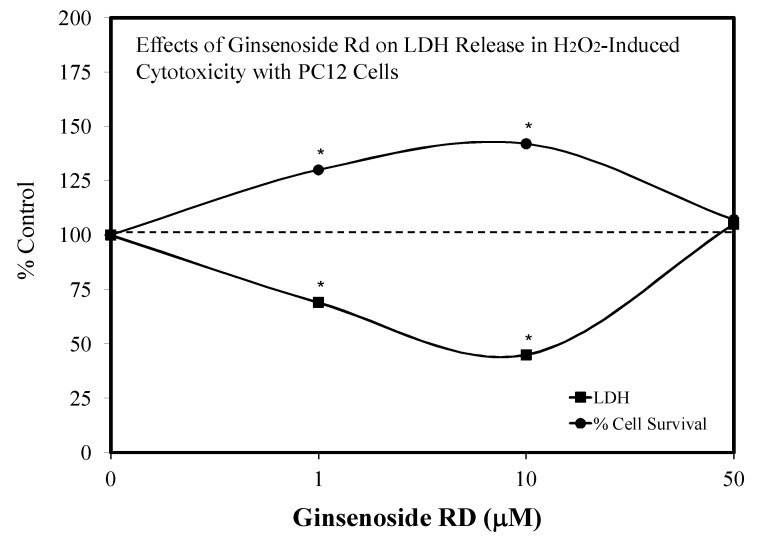
Effects of ginsenoside Rd on LDH release in H2O2-induced cytotoxicity with PC12 cells (source: [102]). Each * represent statistical significance for a data point.

**Figure 25 molecules-25-02719-f025:**
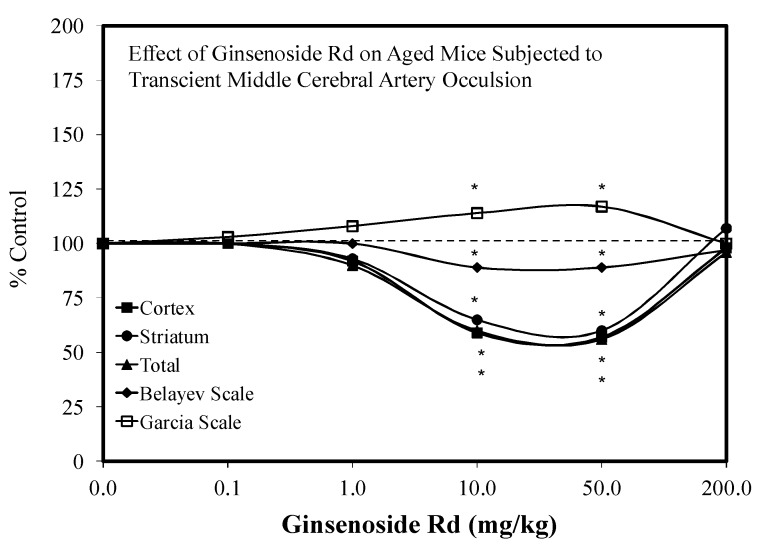
Effect of ginsenoside Rd on aged mice subjected to transcient middle cerebral artery occulsion (source: [103]). Each * represent statistical significance for a data point.

**Figure 26 molecules-25-02719-f026:**
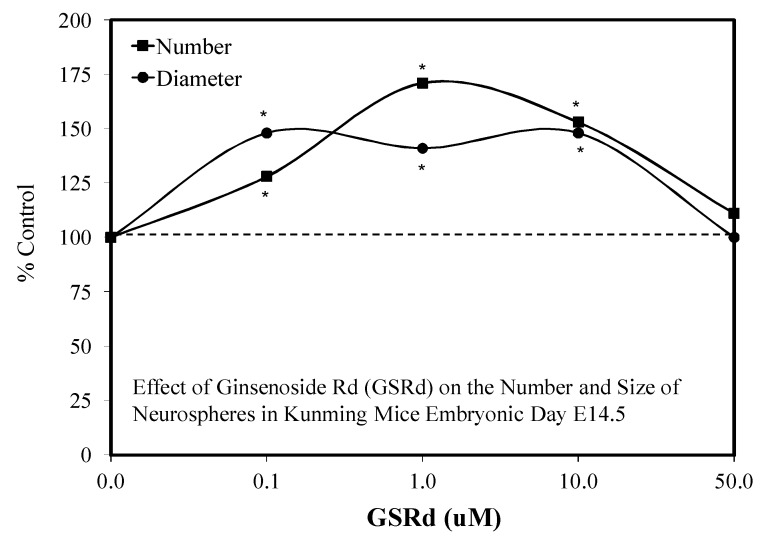
Effect of ginsenoside Rd (GSRd) on the number and size of neurospheres in kunming mice embryonic day E14.5 (Source: [104]). Each * represent statistical significance for a data point.

**Figure 27 molecules-25-02719-f027:**
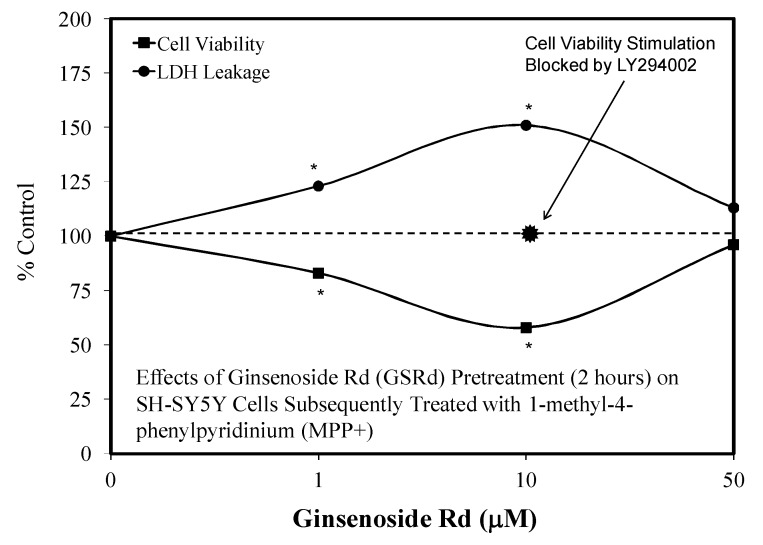
Effects of ginsenoside Rd (GSRd) pretreatment (2 h) on SH-SY5Y cells subsequently treated with 1-methyl-4-phenylpyridinium (MPP+) (source: [105]). Each * represent statistical significance for a data point.

**Figure 28 molecules-25-02719-f028:**
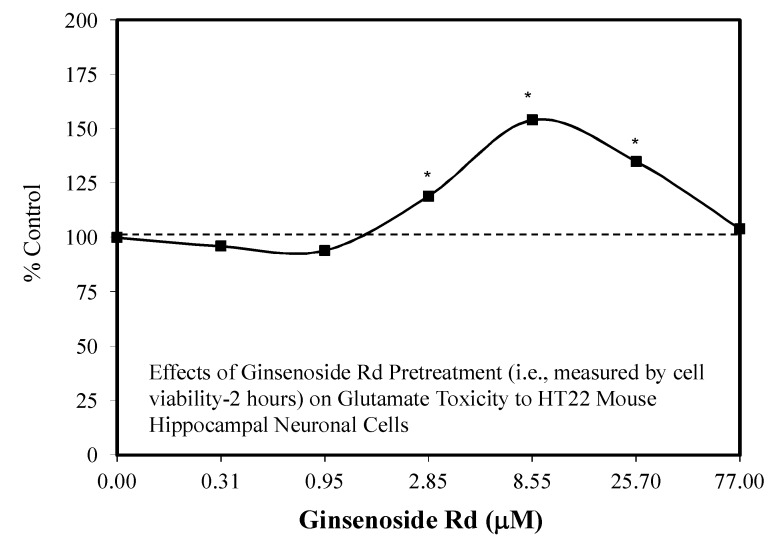
Effects of ginsenoside Rd pretreatment (i.e., measured by cell viability: 2 h) on glutamate toxicity to HT22 mouse hippocampal neuronal cells (source: [106]). Each * represent statistical significance for a data point.

**Figure 29 molecules-25-02719-f029:**
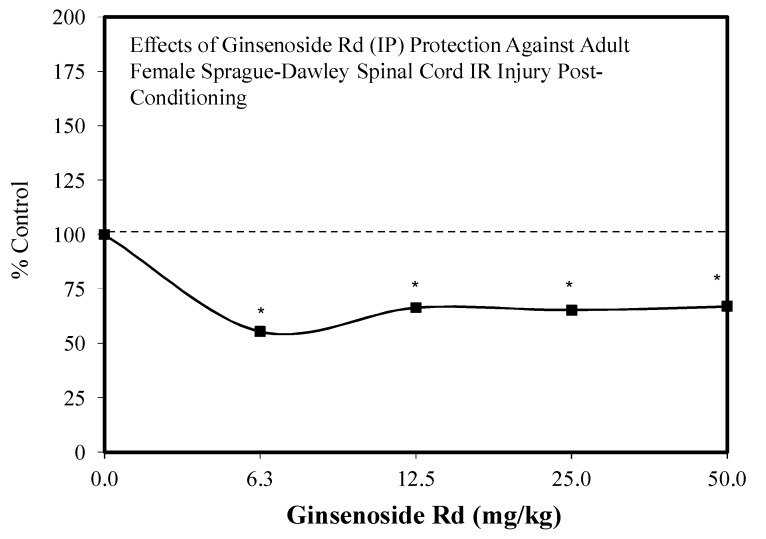
Effects of ginsenoside Rd (IP) protection against adult female sprague-dawley spinal cord IR injury post-conditioning (Source: [107]). Each * represent statistical significance for a data point.

**Figure 30 molecules-25-02719-f030:**
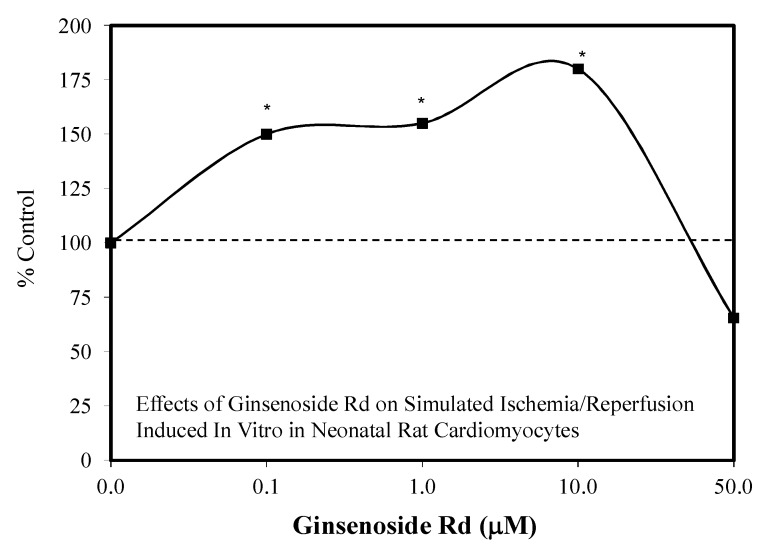
Effects of ginsenoside Rd on simulated ischemia/reperfusion induced in vitro in neonatal rat cardiomyocytes (source: [109]). Each * represent statistical significance for a data point.

**Figure 31 molecules-25-02719-f031:**
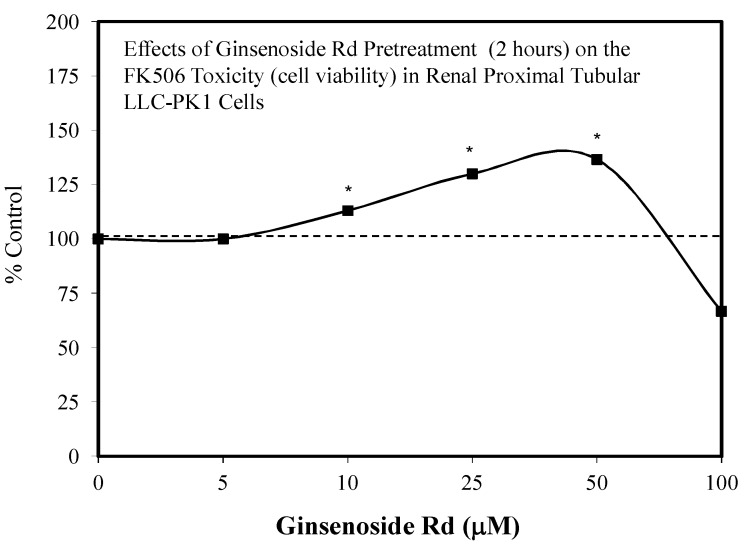
Effects of ginsenoside Rd pretreatment (2 h) on the FK506 toxicity (cell viability) in renal proximal tubular LLC-PK1 cells (Source: [110]). Each * represent statistical significance for a data point.

**Figure 32 molecules-25-02719-f032:**
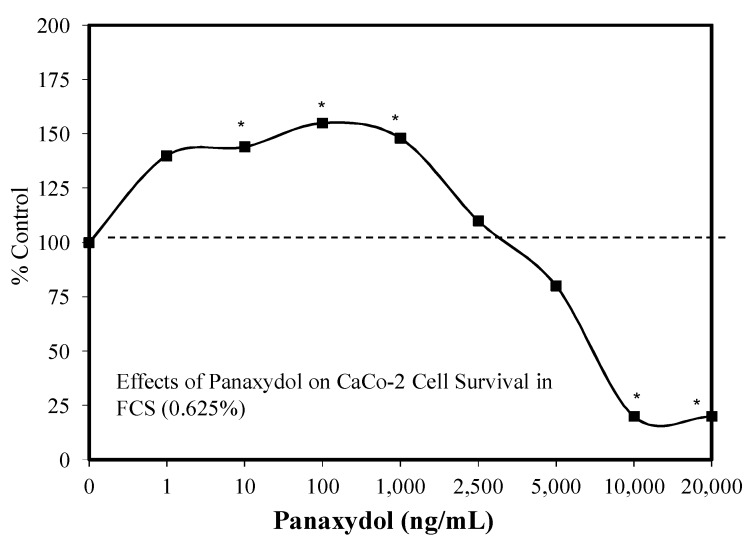
Effects of panaxydol on CaCo-2 cell survival in FCS (0.625%) (Source: [111]). Each * represent statistical significance for a data point.

**Figure 33 molecules-25-02719-f033:**
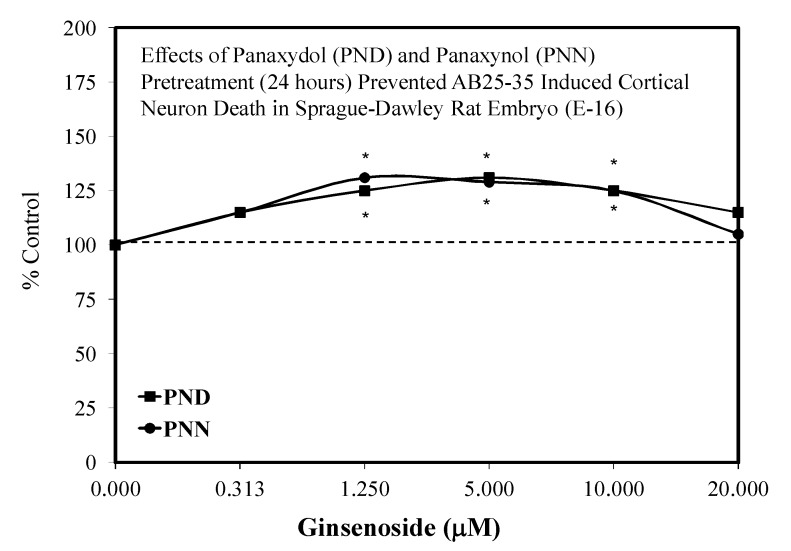
Effects of panaxydol (PND) and panaxynol (PNN) pretreatment (24 h) prevented AB25–35-induced cortical neuron death in Sprague–Dawley rat embryo (E-16) (source: [112]). Each * represent statistical significance for a data point.

**Figure 34 molecules-25-02719-f034:**
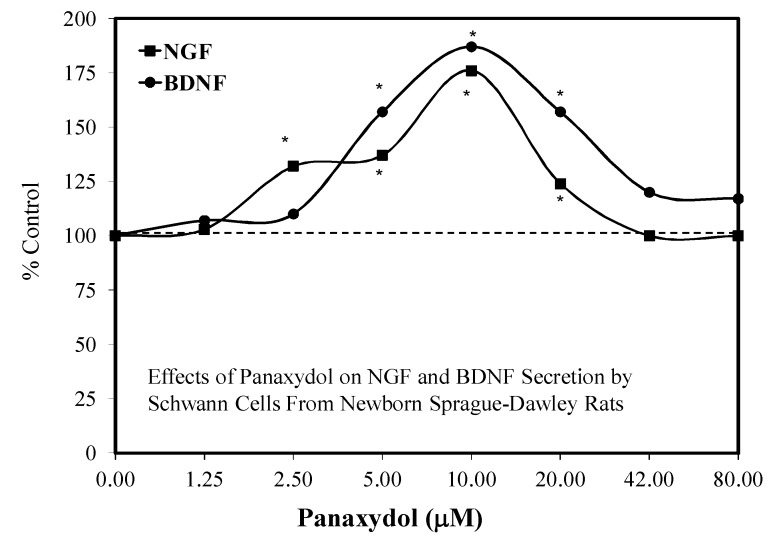
Effects of panaxydol on nerve growth factor (NGF) and brain-derived neurotrophic factor (BDNF) secretion by Schwann cells from newborn Sprague–Dawley rats (Source: [74]). Each * represent statistical significance for a data point.

**Figure 35 molecules-25-02719-f035:**
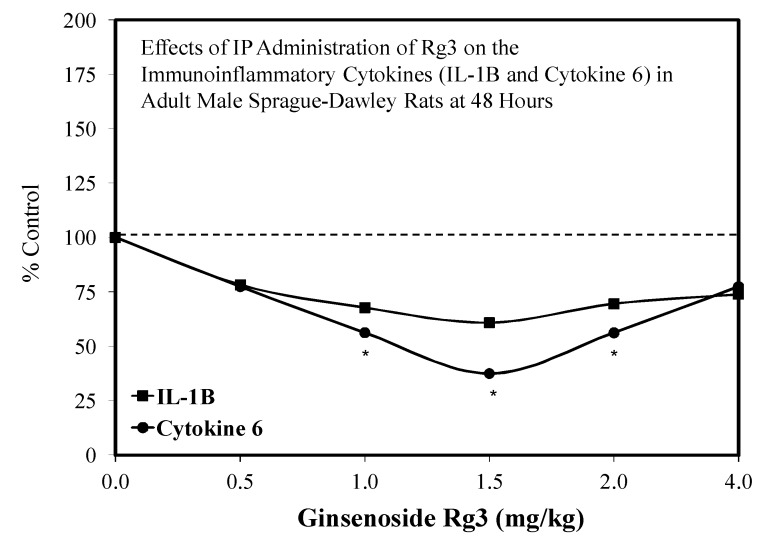
Effects of IP administration of Rg3 on the immunoinflammatory cytokines (IL-1B and cytokine 6) in adult male Sprague–Dawley Rats at 48 hours (source: [113]). Each * represent statistical significance for a data point.

**Figure 36 molecules-25-02719-f036:**
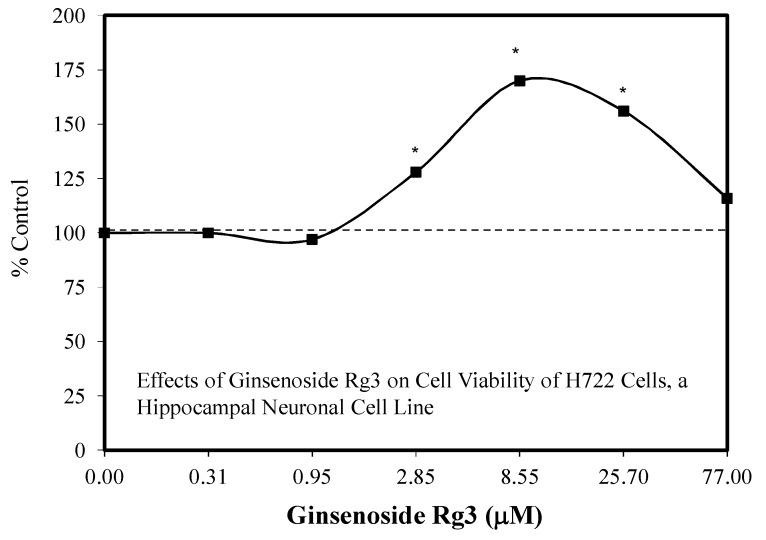
Effects of ginsenoside Rg3 on cell viability of H722 cells, a hippocampal neuronal cell line (source: [106]). Each * represent statistical significance for a data point.

**Figure 37 molecules-25-02719-f037:**
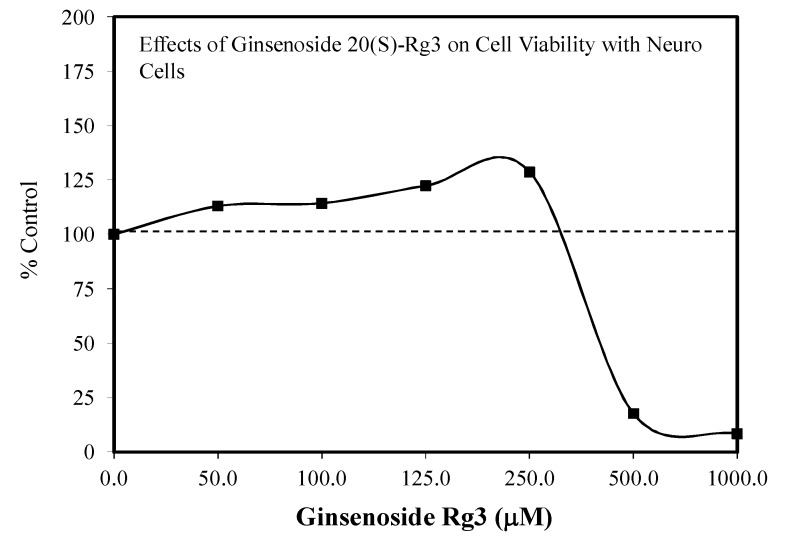
Effects of ginsenoside 20(S)-Rg3 on cell viability with neuro cells (source: [116]).

**Figure 38 molecules-25-02719-f038:**
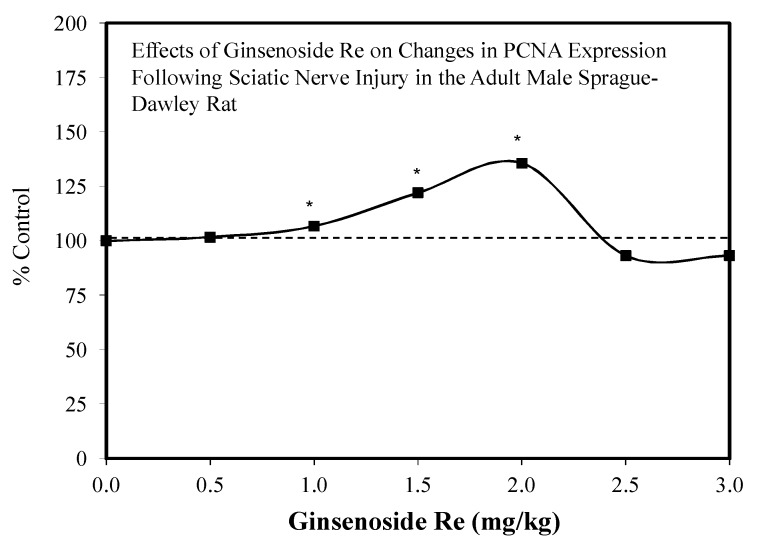
Effects of ginsenoside Re on changes in PCNA expression following sciatic nerve injury in the adult Male Sprague–Dawley rat (source: [120]). Each * represent statistical significance for a data point.

**Figure 39 molecules-25-02719-f039:**
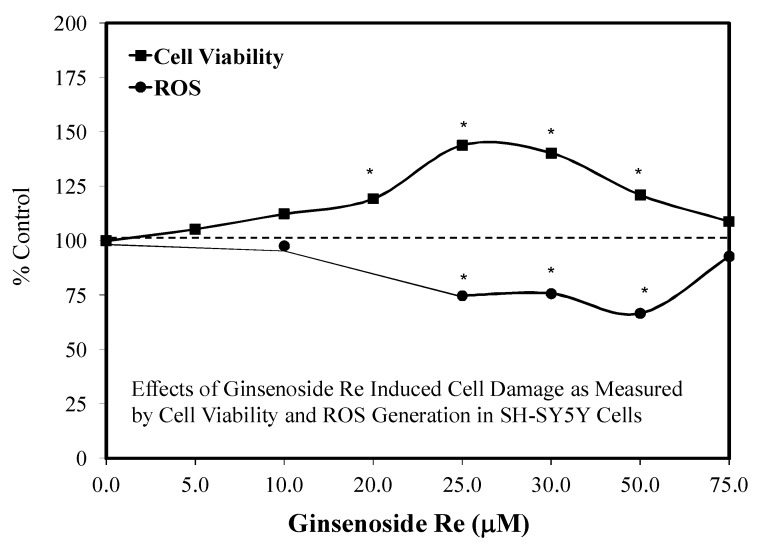
Effects of ginsenoside Re-induced cell damage as measured by cell viability and ROS generation in SH-SY5Y cells (source: [121]). Each * represent statistical significance for a data point.

**Figure 40 molecules-25-02719-f040:**
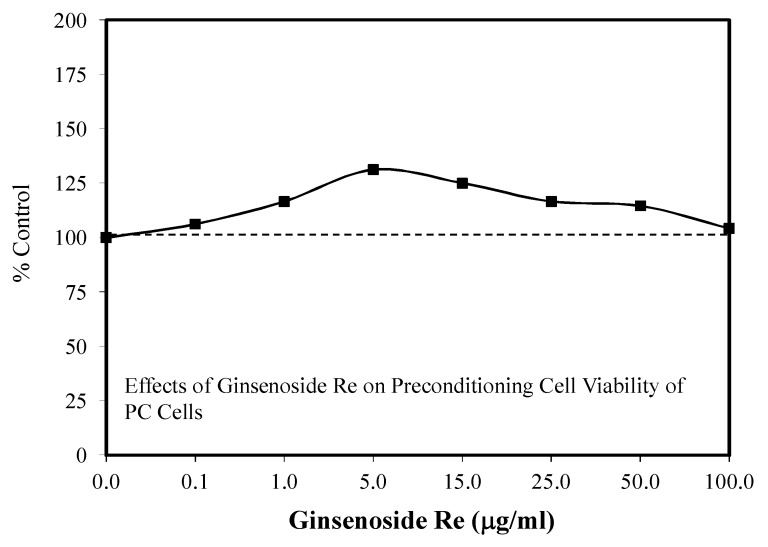
Effects of ginsenoside Re on preconditioning cell viability of PC cells (source: [52]).

**Figure 41 molecules-25-02719-f041:**
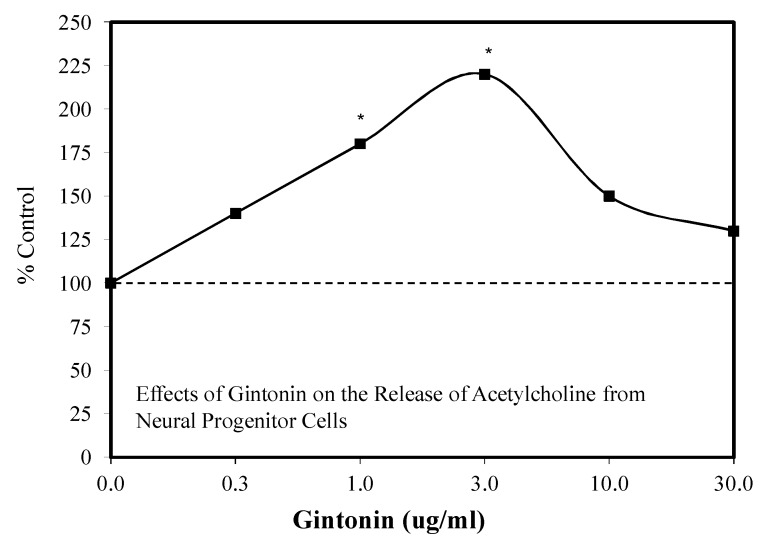
Effects of gintonin on the release of acetylcholine from neural progenitor cells (source: [129,131]). Each * represent statistical significance for a data point.

**Figure 42 molecules-25-02719-f042:**
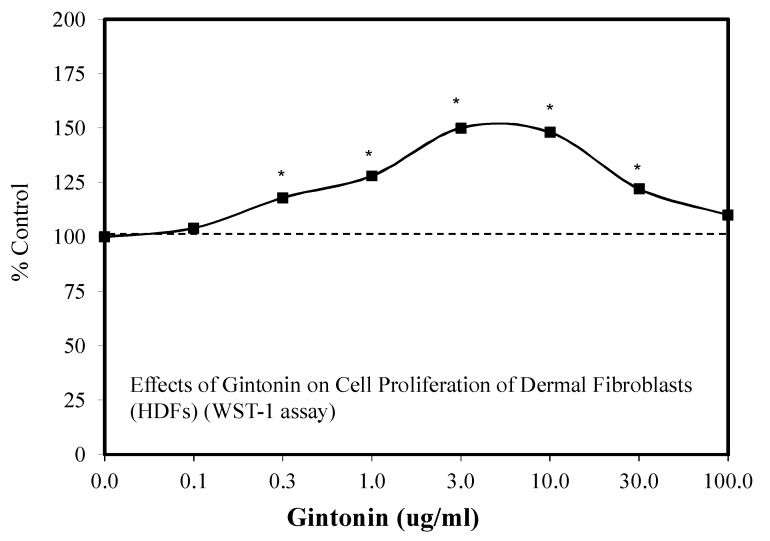
Effects of gintonin on cell proliferation of dermal fibroblasts (HDFs) (WST-1 assay) (source: [123]). Each * represent statistical significance for a data point.

**Figure 43 molecules-25-02719-f043:**
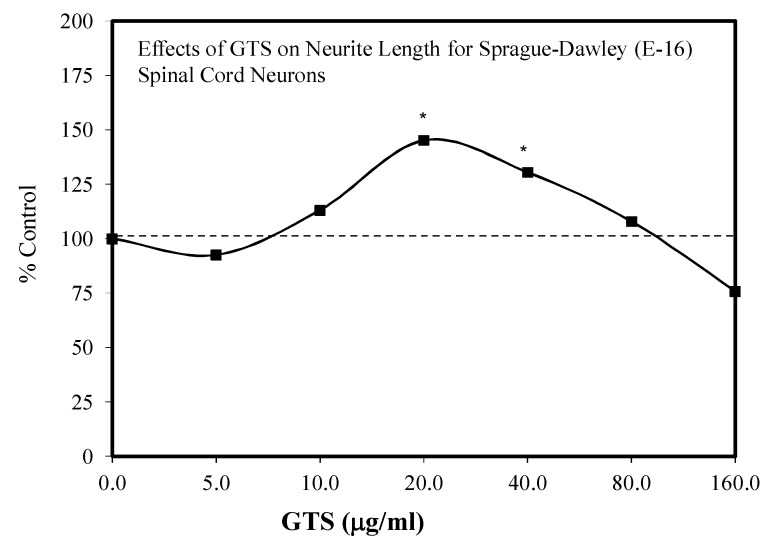
Effects of GTS on neurite length for Sprague–Dawley (E-16) spinal cord neurons (source: [24]). Each * represent statistical significance for a data point.

**Figure 44 molecules-25-02719-f044:**
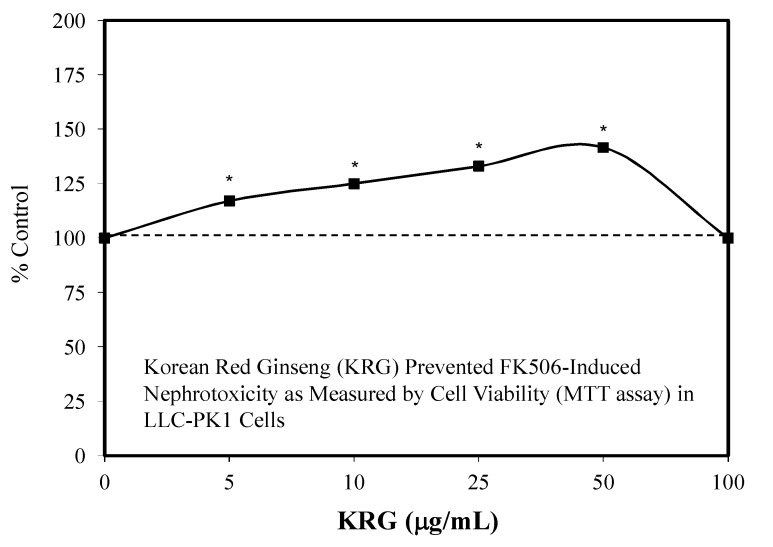
Korean Red Ginseng (KRG) prevented FK506-induced nephrotoxicity as measured by cell viability (MTT assay) in LLC-PK1 cells (source: [133]). Each * represent statistical significance for a data point.

**Figure 45 molecules-25-02719-f045:**
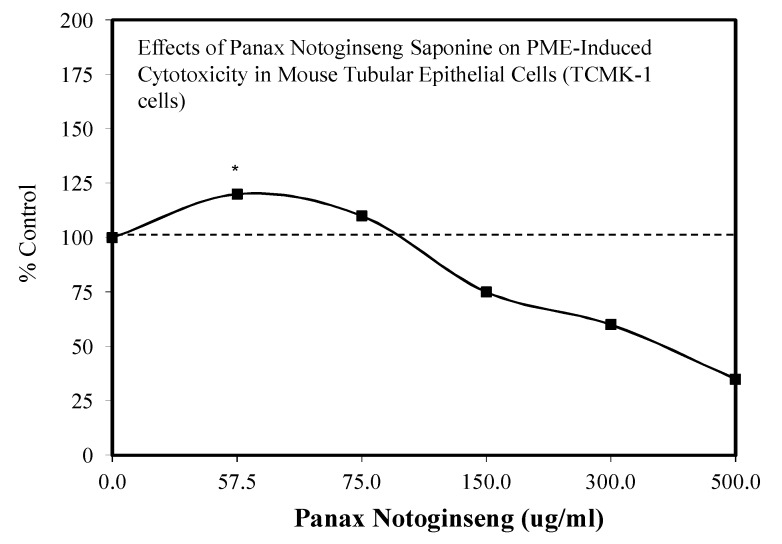
Effects of Panax notoginseng saponine on PME-induced cytotoxicity in mouse tubular epithelial cells (TCMK-1 cells) (source: [144]). Each * represent statistical significance for a data point.

**Figure 46 molecules-25-02719-f046:**
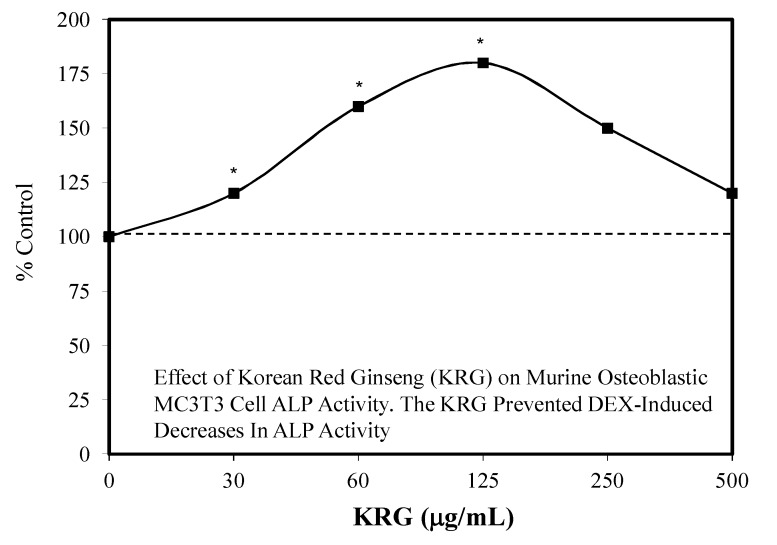
Effect of Korean Red Ginseng (KRG) on murine osteoblastic MC3T3 Cell ALP activity. The KRG prevented DEX-induced decreases in ALP activity (source: [131]). Each * represent statistical significance for a data point.

**Figure 47 molecules-25-02719-f047:**
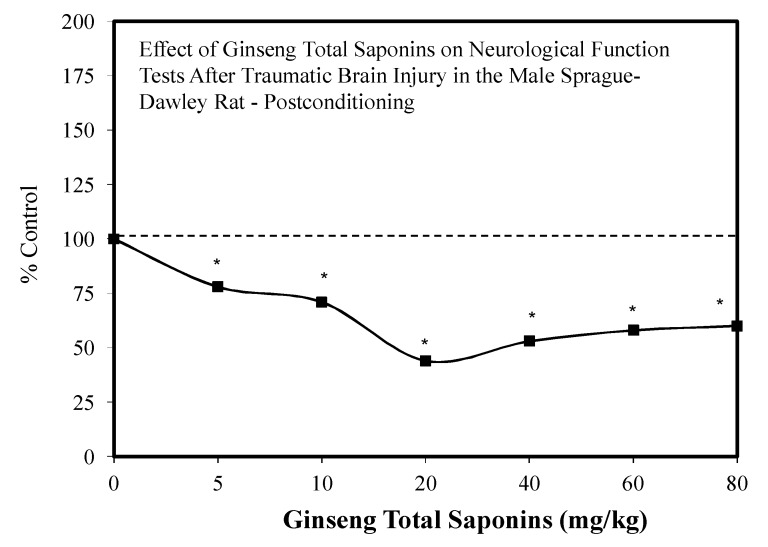
Effect of ginseng total saponins on neurological function tests after traumatic brain injury in the male Sprague–Dawley rat post-conditioning (source: [23]).

**Figure 48 molecules-25-02719-f048:**
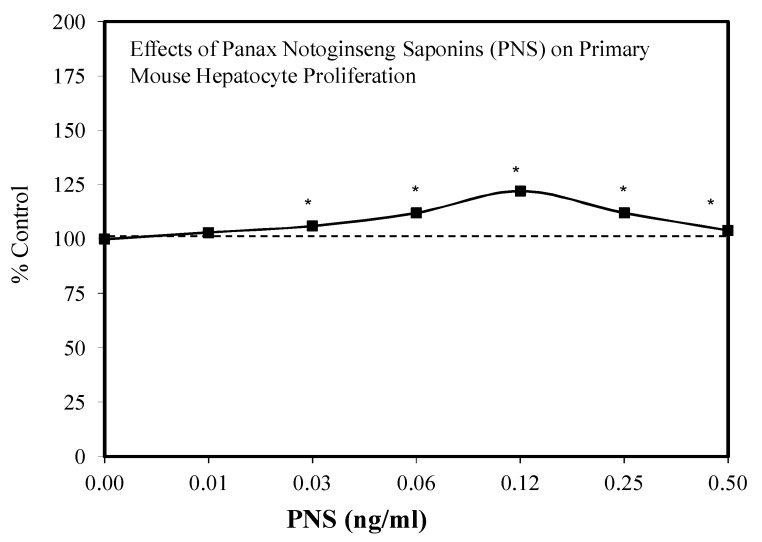
Effects of Panax Notoginseng Saponins (PNS) on primary mouse hepatocyte proliferation (source: [151]).

**Table 1 molecules-25-02719-t001:** Hormetic effect with ginseng mixtures/specific constituents and biological models.

**Non-Neural Models**

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

**Table 2 molecules-25-02719-t002:** Ginseng-induced hormetic dose responses by constituent/mixtures.

Rg1	Rb1	Rd	Rg3	Polyacetylenes	Gentonin	Ginseng Mixtures
Stem cells	Neuroprotection	Neuroprotection	Neuronal injury	Nerve repair	Wound healing	Neuroprotection
Proliferation/differentiation	AD/PD	PD	Neuronal regeneration	AD	Corneal damage	Learning/memory
Neuroprotection	Stroke	Stroke	Atherosclerosis		AD	Behaviors
AD/PD	Neuronal Repair	Neurosphere production			Glycogenolysis	Hippocampal cell
Stroke	Other: Heart, Diabetes	Glutamate toxicity.				Neurite outgrowth
Neonatal brain hypoxia		Spinal Cord				NSC proliferation
Neuronal wound healing		Other: Heart-cardiomyocytes, Kidney-FK506				Brain Injury
Nerve regeneration						Other: Muscle injury, Osteogenesis, Kidney damage, Hormone production
Other: Heart cardiomyocytes, Immune enhancement, Diabetes						

**Table 3 molecules-25-02719-t003:** Ginseng-induced hormetic dose response disease and endpoint.

Alzheimer′s Disease Prevention: Rg1
Parkinson’s Disease Prevention: Rg1
Stroke Damage Prevention: Rg1
Kidney Damage Prevention: Rd
Heart Related Damage Prevention: Rg1
Nerve Cell Damage Prevention/Regeneration: Rg1
Diabetes Prevention: Rg1, Rb1, Rg3
Stem Cell Proliferation/Differential: Rg1 and mixtures.
Brain Traumatic Injury: Mixtures

**Table 4 molecules-25-02719-t004:** Hormesis dose responses (median values -% stimulation).

	Mixtures	Rb1	Rd	Rg1	Gintonin	Re
Max Stim	155.5	169.6	146/59.5 (J)	180	172.5	133
Stim Range	7	53.3	10	10	20	25
Sample Size	29	26	6	50	8	7

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
