# Peer review of "Hormesis and Ginseng: Ginseng Mixtures and Individual Constituents Commonly Display Hormesis Dose Responses, Especially for Neuroprotective Effects"

_molecules, 2020, doi:10.3390/molecules25112719_

Round 1

Reviewer 1 Report

This is a well written review on hormesis and ginseng. The author has done a great job in providing a comprehensive list of relevant publications. The information provided by this review should provide readers a comprehensive perspective on this topic. There are a several suggestions that could be taken to strengthen the review:

  1. This is a lengthily article. Some of the evidence/text could be summarized in table format. A schematic diagram may be used to summarize the hormesis action of ginseng.
  2. The evidence for the hormesis action of ginseng is based primarily on the stimulatory phase as data for the secondary inhibitory phase in most cases are not available. Some of the truly biphasic responses shown were related to cell proliferation and viability, which may not truly reflective of hormesis. As with any substances, cellular toxicity or cell death will be induced at high concentration in non-specific manner. The author has pointed out this deficiency in some of the cases; but perhaps this aspect should be discussed as a whole.  Furthermore, is this observation common to other substances that exhibit hormesis?
  3. The discussion with reference to individual ginsenosides is fine from a mechanistic perspective. However, the extension to the effect of intake of ginseng in humans following oral administration is problematic. The context of the ginseng mixtures cited should be examined and reviewed more carefully with reference to the difference types or species of ginseng. The two major species are the Panax ginseng and Panax quinquefolius, which as dissimilar ginsenoside profile, while notoginseng is a minor ginseng species with different phytochemical profile. Korean red ginseng is derived from processing of Panax ginseng by steaming, generating Rg3 and other more lipophilic ginsenosides.
  4. The attempt to discuss the relevance of in vitro studies with individual ginsenosides to in vivo situation taking in the pharmacokinetic consideration is appropriate but there are challenges that needs to be addressed. The role of the microbiome in the intestinal metabolism of ginsenosides should be taken into consideration. Thi may be relevant to the examination of data derived from administration by oral Vs IP injection.
  5. The author has attempted to discuss the relevance of experimental studies in vitro and in vivo to the human situation. Please note that because of the difference in rate of metabolism and body surface, the effective dose of most drugs should be 5 times higher in rodents as compared with humans.

Author Response

General:

I would like to thank both reviewers for their detailed reading of the manuscript on Hormesis and ginseng.  Your considerable efforts are greatly appreciated.

Reviewer #1

  1. Much thought was given to the idea of the reviewer during the preparation of this manuscript.  I also had several “friendly” detailed reviews by colleagues prior to submittal of this manuscript and this general question was posed.  In the end, I developed Table 2 which provides an organizational type summary table.  Since the findings reviewed here are extensive this seemed to be a helpful table that provides need directional focus.
  2. The issue raised by the reviewer would be good to better address in this manuscript.  Consequently I have added this issue in the discussion.  Thanks for this suggestion.
  3. I appreciate the issue raised by the reviewer about ginseng mixtures difference in individual constituents. While this issue was address in the ginseng mixture section to the extent that seemed appropriate and placed in proper biological and public health context.. I don’t believe that more assessment on this topic is needed.
  4. The issue of pharmacokinetics of individual ginsenosides is an important area.  However, this is beyond the general scope of this manuscript.
  5. 5. No comment is needed.

Reviewer 2 Report

The study is very interesting. 

In this rewier  Edward J. Calabrese describes the ability of  ginseng mixtures and individual ginseng chemical constituents to modulate and to induce hormetic dose responses in numerous biological models.

This study is well done and exhaustive. Interestingly the author report the studies on the main costituent of ginseng (RG1, RB1) and  principal ginseng mixtures. The range of endpoints assessed has  been broad but the greater interest of the author concerns the area of neuroprotection, in particular cellular/animal model  for AD and PD disease, followed by stroke, neuronal health/regeneration based on studies with spinal cord /sciatic nerve,  injuries and pain. He provides the large number of hormetic dose response observations in the experimental studies.

For all these considerations the work can be accepted in the following form.

Author Response

General:

I would like to thank both reviewers for their detailed reading of the manuscript on Hormesis and ginseng.  Your considerable efforts are greatly appreciated.

Reviewer # 2

No comments are needed.